# Coherent poly propagation materials with 3-dimensional photonic control over visible light

Michelle R. Stem  *

Department of Materials Research, Complete Consulting Services, LLC., Big Rapids, Michigan, United States of America

* MRStem@CompleteConsultingServices.com

**Data Availability Statement:** All relevant data are within the paper and its Supporting Information files.

**Funding:** Complete Consulting Services, LLC. provided support in the form of access to research

## Abstract

The purpose of the present research was to identify and examine materials demonstrating a previously undiscovered property of coherent poly propagation (CPP). The materials were amorphous silicates as natural precious opals. CPP enabled three-dimensional photonic control over mono and polychromatic visible light wavelengths. CPP caused coherent diffraction of incident poly and monochromatic light. Apart from the iconic play-of-color of precious opal, CPP specimens demonstrated diffractive photonic demultiplexing and/or upconversion and/or downconversion of incident light with strong photonic coherence such that the shape of the incident light source was propagated over three dimensions over multiple visible frequencies. CPP events manifested as each specimen was rocked under the incident light. Additionally, the specimens demonstrated atypical control over internally reflected and transmitted light. The specimens applied axial rotational symmetry over the incident light. Amorphous materials would be expected to exert no symmetry control. CPP and rotational properties occurred in isolation from exogenous thermal, photonic and electrical influences. Furthermore, several non-destructive analytical instruments were employed, such as: spectrophotometer, polariscope and refractometer. The analytical methods revealed unusual behaviors of these specimens. The application of materials demonstrating three-dimensional photonic control will have far-reaching implications for many industries, including: photonic wavelength demultiplexing, fiber optics, imaging, microscopy, projections, security, cryptography, computers and communications.

## Introduction

Natural precious opals are comprised, mostly, of nano-sized silicate spheres and adsorbed $H_2O$. A silicate matrix ($SiO_2$:$H_2O$) has variably sized spheres with no play-of-color [1]. In this matrix are embedded pseudo-crystalline zones (PCZs) with uniformly sized, highly ordered spheres of variable overall dimensions [1,2]. PCZ boundaries, depths and orientations vary. Each PCZ diffracts incident light independently of the other PCZs, causing opal's iconic play-of-color. Precious opal is like an amorphous sea of non-uniform hydrated silicate nano-spheres, embedded with islands of PCZs [1,3–6].

equipment for MRS with no other financial, material contributions or considerations. The funder provided support in the form of access to research equipment only, but did not have any additional role in the study design, data collection and analysis, decision to publish, or preparation of the manuscript. The specific roles of the author is articulated in the 'author contributions' section.

**Competing interests:** I have read the journal's policy and have the following competing interests: Complete Consulting Services, LLC. provided support in the form of access to research equipment for MRS with no other financial, material contributions or considerations. There are no patents, products in development or marketed products associated with this research to declare. This does not alter our adherence to PLOS ONE policies on sharing data and materials.

Opal has been a template for many years for photonic control materials. Strides have been made towards two-dimensional (2-d) photonic control [7–11]. Additionally, research for a material that allows three-dimensional (3-d) photonic control has been ongoing for many years [12–15]. The present research has identified several specimens that demonstrated 3-d photonic control over the visible spectrum.

The purpose of the present research was to identify and examine eight specimens of precious opal that allowed three-dimensional visible light photonic control via the previously unidentified property of coherent poly propagation (CPP) [16,17]. CPP allowed this researcher to exert a specialized form of photonic demultiplexing [18,19]. The specimens diffracted incoming visible light and accurately propagated the shapes of incident photon sources over some or all of each diffracted frequency, behaving as a new form of demultiplexing photonic waveguide (Figs 1 and 2). The propagated shapes were able to be manually moved on the curved surfaces of the three dimensional specimens. Also, the research specimens exerted a previously unknown photonic rotational symmetry operation.

The development of CPP materials may have applications in many fields. The symmetry operation may allow a photonic means to generate computational yes/no or one/zero signals based on the orientation of the image which contrasts to the current methods which are electron limited. Security, communications, cryptography and imaging may be affected by new methods, such as: multiple simultaneous wavelength transmissions (simulpathing) of tamper sensitive data, non-repudiation through selective wavelength masking and water-marking of image transmissions by adding or deleting specific wavelengths. Projections and defense may be affected by the ability to create real-time false ghost projections of high value assets, such as: military planes in-flight, drones, ground-based and other assets. Computers may be affected by the simultaneous calculation/verification of critical data in a multi-optical processor environment, corresponding to a photonic version of the multi-electron-based processor systems currently deployed. Microscopy may be affected by enabling real-time simultaneous imaging over multiple wavelengths without the time-delay and computational vulnerability of current image processing. The CPP property and symmetry operation may affect fiber optics. Current photonic wavelength demultiplexing requires poly-frequency photons be transitioned to and diffracted by intermediate materials where the fiber optic system loses photon density and signal integrity. The CPP property may be developed to be deployed in-line to lessen such loss via

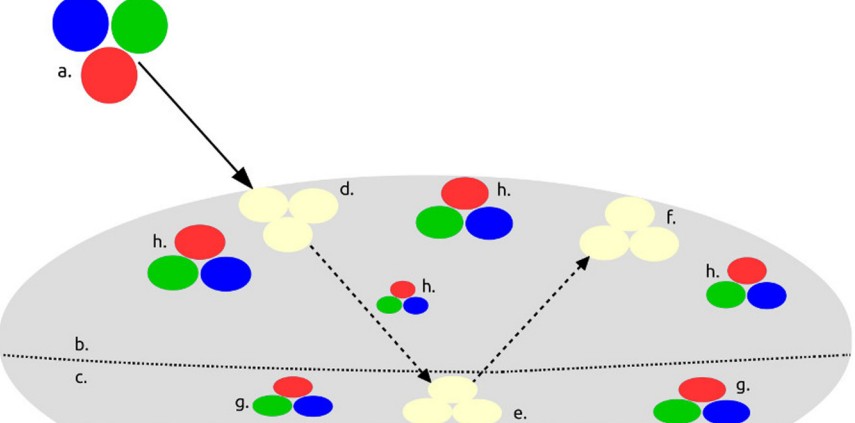

**Fig 1. Graphical depiction of CPP and PPOI with axial rotation.** a. incident light, b. proximal side, c. distal side, d. incident PPOI, e. rotated transmitted PPOI, f. rotated reflected PPOI, g. rotated transmitted CPP, h. rotated reflected CPP. Dotted lines depict photon flow and demarcation between proximal (top) and distal (underside) sides.

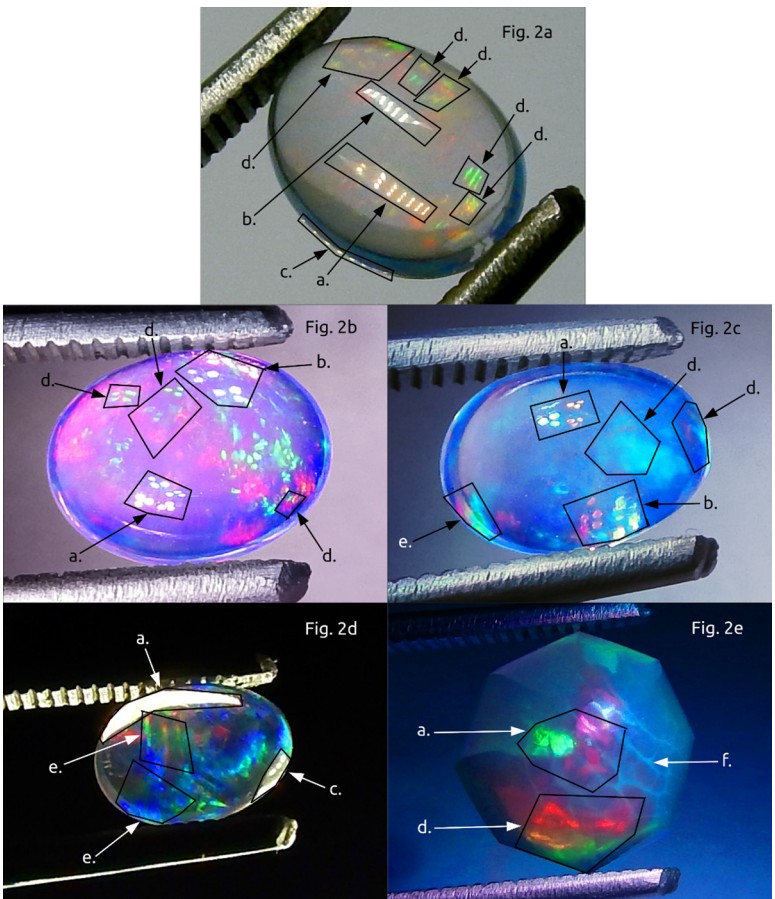

**Fig 2. Close-up photographic primer to help readers identify CPP and PPOI events discussed in this article.** a. incident PPOI, b. rotated reflected PPOI, c. rotated transmitted PPOI, d. rotated reflected CPP, e. rotated transmitted CPP, f. photonic glass borders of macroscopic play-of-color zones. The incident light for Fig. b, c, and e are a cluster of twelve fiber optic cables, three each in red, yellow, green and blue. The incident light for Fig. a and d are a lamp with multiple rows of three white light LEDs. Fig c and d show the distal side and fig a, b, c and e show the proximal side. CPP and PPOI events assume the incident shape, but at axially rotated orientations after the incident PPOI. Generally, the CPP property propagates multiple events in a specimen. While CPP events sometimes propagate the incident colors, they also upconvert and/or downconvert incident colors across the visible spectrum. Conversion also occurs as CPP events glide over the surface of a specimen while rotated under the incident light. Contrast and saturation have been enhanced for these photos to facilitate visual identification of the photonic properties. In the research, Fig. c and d are specimen 2, Fig. a and b are specimen 4 and Fig. e is specimen 7.

the creation of multi-cast capable fiber switches and routers that do not require an electron conversion step. Furthermore, application of CPP to fiber optic cables may enable rotational shifting of transmitted fiber optic data with wavelength alteration. As stated, further development of CPP materials with 3-d photonic control will have far-reaching, positive and disruptive implications for many industries and fields, such as: photonic wavelength demultiplexing, fiber optics, imaging, microscopy, projections, security, cryptography, computers and communications [18,19,20–35].

## Materials and methods

### General

The present research examined eight specimens of naturally occurring, untreated precious opal (Table 1) from two geographical regions. Five of the specimens (3, 5, 6, 7 and 8) originated

from mines in the Welo region of Ethiopia (11˚ 20' N, 40˚ 00' E) and three of the specimens (1, 2 and 4) originated from mines in the Magdalena region of Mexico (20˚ 55' N, 103˚ 57' W). All specimens were obtained by the author via retail purchases in the USA. The author was not involved in the mining process for any of the specimens. It is not possible to provide further specific geographical coordinates for the mines because the Welo Ethiopia and Magdalena Mexico areas have numerous mines operated by various sized operators. Identifying the specific mines from which each specimen originated may not be possible because there are likely to be no consistent chemical differences between the various mines in each of the two areas. Even so, determination would require additional funding, substantial numbers of specimens and destructive testing.

Untreated specimens were not dyed, coated, heated, infused, sealed, irradiated, or made in a laboratory. All specimens were shaped into cabochons with one specimen partially faceted. None of the specimens emitted alpha or beta particles in detectable levels and the only chemicals they were exposed to were $H_2O(l)$ and ambient air. In various light sources, the specimens displayed mild to intense play-of-color and contra luz. Seven specimens were colorless transparent and one specimen was orange transparent. The specimens were viewed from the same side onto which the light was initially incident (proximal) and from the side opposite to the one impacted by the incident light (distal).

## Hydration

To maximize reproducibility, this research kept each specimen at maximum hydration [1,4] prior to conducting each step in analysis. Changes in internal hydration levels could have affected the measured and computed values [36,37]. Hence, before each research step, each specimen was submerged in individual containers of purified $H_2O(l)$ for at least twenty four hours and then dried with a Kim-Wipe until dry to the touch (a few seconds). Between analyses, each specimen was re-submerged in $H_2O(l)$.

## Ambient incident photonic and electronic control

Since this research focused on photonic interactions and control, each phase of the present research was considered to be photonically and electronically sensitive [1,4]. Hence, extraneous photons and electrons in the laboratory were severely minimized to maximize the reliability/repeatability of observations and enhance the detection of properties which may be sensitive to, or affected by, stray photons. Minimal ambient light in the laboratory was established by closing all doors, closing all window coverings, covering windows with light blocking materials, shutting off or covering all undesired nearby photon sources and conducting observations after astronomical twilight. Yet, observations did not occur under conditions of total darkness because one closed door was not covered with light-blocking materials and three LED lights from switches of in-use laboratory equipment in the room were blocked, not extinguished. The door and LEDs were located indirectly and at least three meters from the experimental set-up. Minimal ambient light was defined as a density of photons that was too low to register on the colorimeter in any visible frequency (Table 1). All electrical cables or potential sources of static electricity were either disconnected, grounded or damped magnetically [4].

In addition to ambient light control, steps were taken to ensure that experiment observations were limited to responses due to the internal properties of the specimens and not due to other surfaces, reflections or some other exogenous source [1,4]. Hence, for macroscopic observations, each specimen was examined while held aloft in an articulating, photonically unresponsive holder at approximately 35 mm above a photonically unresponsive surface of neutral light beige color. For microscopic observations, each specimen was examined on a

**Table 1. Materials and equipment.**

| Materials and Equipment | Description |
|---|---|
| Laser | 650 nm, <5 mW, constant wave |
| Laser | 593.5 nm, <5 mW, constant wave |
| Laser | 532 nm, <5 mW, constant wave |
| Laser | 450 nm, <5 mW, constant wave |
| Laser | 405 nm, <5 mW, constant wave |
| Long wave ultraviolet (LWUV) | 375 nm, visible light quartz filter, 6 W Hg fluorescent bulb |
| Mid wave ultraviolet (MWUV) | 307 nm, visible light quartz filter, 6 W Hg fluorescent bulb |
| Short wave ultraviolet (SWUV) | 254 nm, visible light quartz filter, 6 W Hg fluorescent bulb |
| Polychromatic light source, LED | 6500 K (2000–6500 K), CRI > 90, 14 W, IR and UV filters, TaoTronics Elune LED desk lamp, model TTDL02 |
| Polychromatic and monochromatic light source, fiber optic | 6000 K, CRI = 90, 75 W, DMX, Chinly |
| Polychromatic light source, compact fluorescent | 5500 K, CRI = 90, 23 W, full spectrum, generic articulated arm desk lamp |
| Polychromatic light source, halogen | 3200 K, CRI = 100, 15 W, microscope stage light |
| Microscope | Amscope, model SM-2TYY, magnification 4–180 |
| Camera, microscope | Amscope, 10 MP, model MW1000-ck |
| Camera, macro | Nikon Coolpix L820 |
| Colorimeter | Sekonic C-700-U, MPN 401–702, CMOS linear image sensor, λ = 380–780 nm |
| Infrared (IR) thermal imager | Range: -50–380 ˚C |
| Mass balance | GemPro digital, range: 0.025–125.000 g |
| Spectrophotometer | Single beam, range: λ = 320–1100 nm |
| Refractometer | Sinotech digital, range: 1.300–2.100 |
| Thermal conduction analyzer | Presidium |
| Dichroscope | Calcite |
| Polariscope | Tabletop model |

photonically unresponsive white microscopy plate. All lamps, holders, materials and clothing were tested to ensure that none were photonically responsive to any of the wavelengths being utilized. Not photonically responsive meant that each component of the set-up had minimal reflectivity and did not display, or cause any specimens to display, photonic reactivity to any of the wavelengths applied (e.g. no color change, fluorescence, phosphorescence, or reflected colors).

## Incident light sources

A series of sixteen different photon sources were made incident upon each specimen, including: four visible light polychromatic, four monochromatic visible light sources, three ultraviolet (UV) frequencies and five visible light laser frequencies (Table 1) (Fig 3). The purpose of these exposures was to identify and examine the photonic properties and behaviors of each research specimen relative to the CPP property. Each of the non-laser visible light sources was analyzed via colorimetry (Fig 4). However, a colorimetric scan of the blue fiber optic light source was not possible because the colorimetric temperature of the blue light exceeded the analytical capability of the colorimeter. Observations with most of the photon sources were conducted by varying the photonic angles of incidence, relative to the specimen and viewer. Only laser sources were observed with the specimens in a fixed position relative to the incident

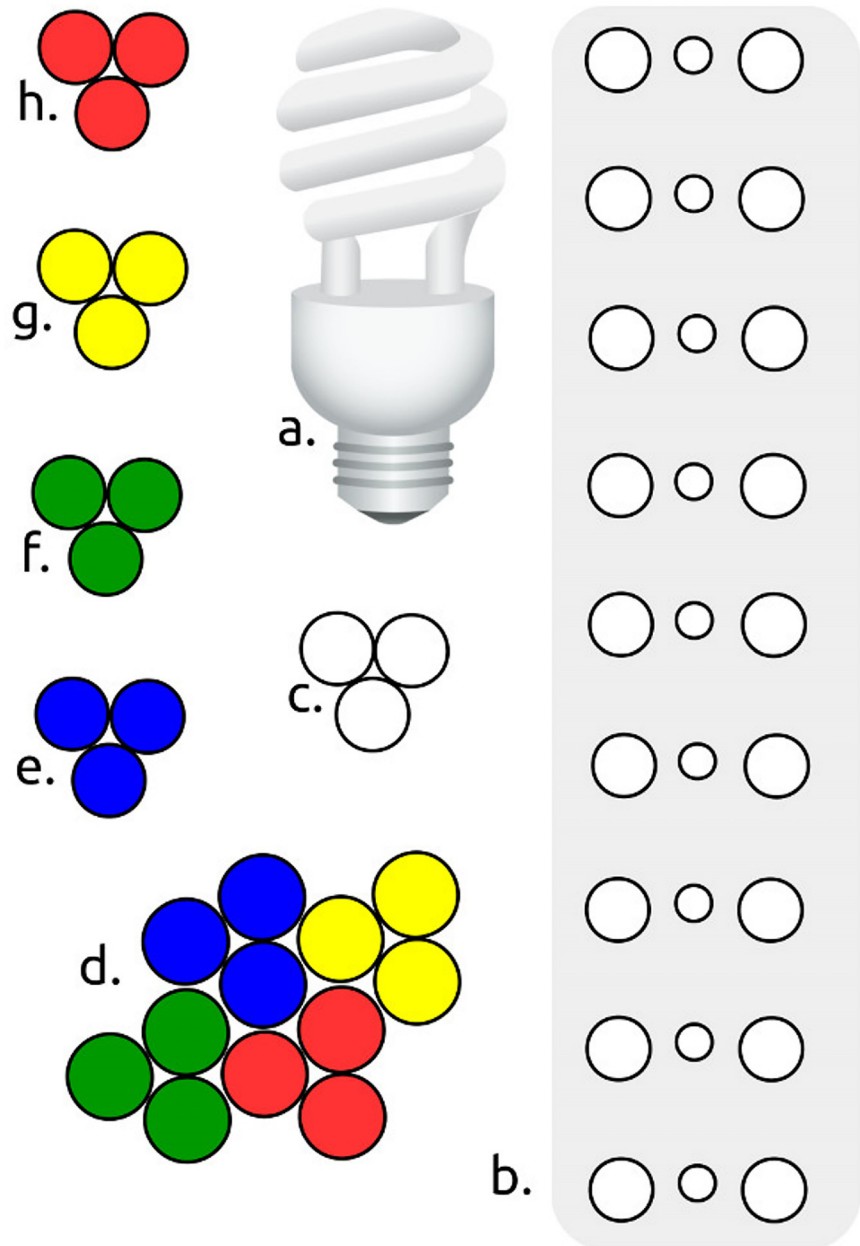

**Fig 3. Diagrams of visible light bulb configurations (not to scale).** a. CFL, white, b. LED, white, c. Fiber optic, white, d. Fiber optic, quad-chromatic, e. Fiber optic, blue, f. Fiber optic, green, g. Fiber optic, yellow, h. Fiber optic, red.

lasers via careful aim of the incident photonic point of impact (PPOI) (Fig 1). The specimens were observed macroscopically and microscopically while exposed to each photon source.

Of the visible light sources, six of them were fiber optic cables (Table 1) (Fig 3). The fiber optic research set-up consisted of clusters of fiber optic cables that transmitted two polychromatic and four monochromatic light sources. A cluster of three cables of unfiltered polychromatic light was made incident on each specimen. Using the same light source, four clusters, each of three cables were prepared with the transmissive ends of each of the four clusters of

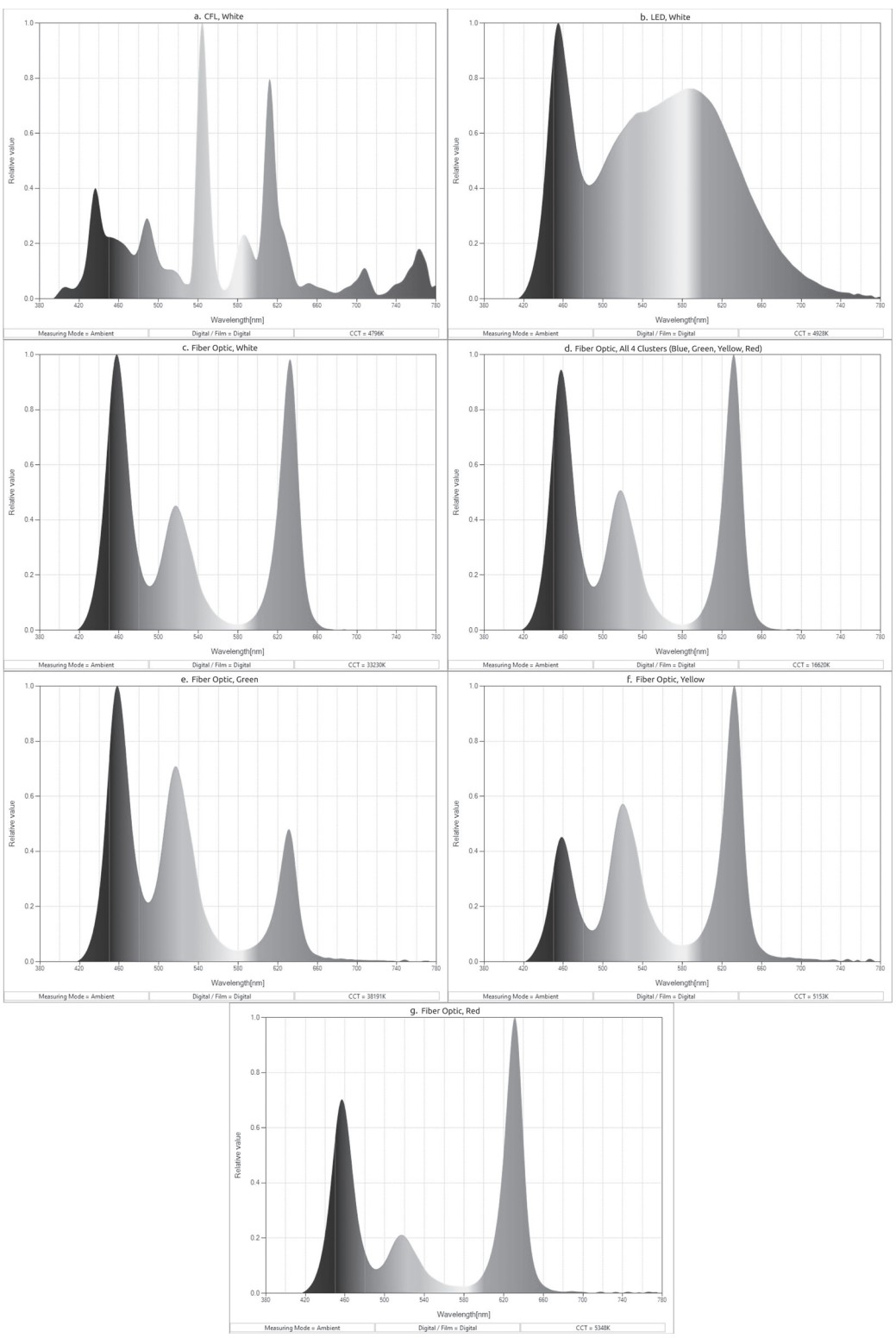

**Fig 4. Colorimetric scans of almost all of the visible light sources.** The colorimetric scan of blue fiber optic was outside of instrumentation limits. a. CFL white, b. LED white, c. fiber optic white, d. fiber optic quad-chromatic, e. fiber optic green, f. fiber optic yellow, g. fiber optic red.

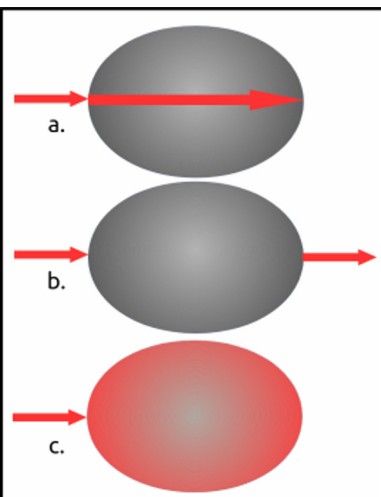

**Fig 5. Diagram of laser direction and interactions.** a. Coherent laser traversal of body (visible path), b. Distal coherent exit of laser (transmission), c. Body glow, texture, color.

three cables dyed monochromatically with archival quality, acid free pigment inks (one cluster each of blue, green, yellow and red). One other polychromatic source was a twelve cable bundle of all four of the monochromatic fiber optic clusters. Thus, a range of six mono and polychromatic fiber optic clusters was made incident on each specimen. A series of five laser sources were made incident upon each of the specimens (Table 1) (Fig 3). Because the external housing diameters of each laser was less than 10 mm, each laser was small enough to be held in a laser lamp holder. The holder allowed each laser to be applied with the needed motionlessness. To further avoid misalignments and/or vibrations, each laser was activated via a remote on/off switch. Further minimization of vibration was accomplished with anti-vibration insulation and halted movement in the building. Each laser was aligned to intersect with each specimen long-wise (Fig 5).

A series of three UV photon sources were made incident on each of the specimens (Table 1) [1,4,5]. Because the external housing of each of the UV source lamps was comparable to a large flashlight, each lamp was too large to be secured in a standard laser lamp holder. Since vibration was not a significant concern with these light sources, each lamp was secured to a ring-stand with clamps and hand-held. Examination of the specimens occurred with the UV sources kept stationary (ring-stand) while the specimens were made mobile in an articulated holder and while the specimens were kept stationary while the UV sources were made mobile (hand-held).

### Spectrophotometer

Solid-state spectrophotometric analytical scans were conducted on each of the specimens over 320 to 1100 nm (Table 1). Each specimen had four spectrophotometric scans, with the specimen placed in a series of four different orientations for each scan: top to bottom, bottom to top, side to side narrow-wise, and side to side long-wise (Fig 6). For each specimen, the top was the dome, the bottom was the flat side opposite the top, and narrow-wise to long-wise were profile views of the specimens that were perpendicular to each other. These analyses showed the levels of photonic absorption of each specimen when exposed to a range of the spectrophotometric wavelengths.

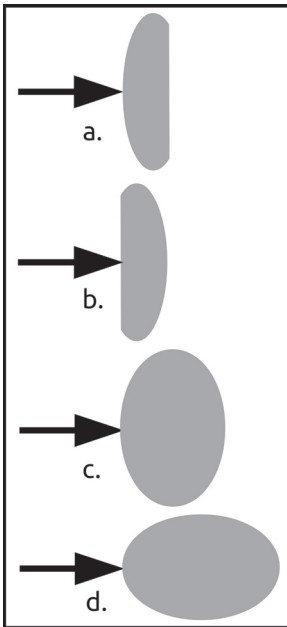

**Fig 6. Diagram of spectrophotometric scan directions.** a. top to bottom, b. bottom to top, c. side-to-side narrow, d. side-to-side long.

## Refractive index

Refractive index (RI) analysis was conducted on each specimen (Table 1). RI is a common test to help identify the material composition of minerals [1,5,6,38,39]. In addition to confirming the mineral species of the specimens, RI was used to try and ascertain whether the PCZs of any of the specimens would alter the specimen RI response to more closely resemble that for crystalline silicates.

## Thermal conductivity

The thermal conductivity of each specimen was tested (Table 1). Thermal conductivity is a common test to discern between many types of minerals [1,38,40]. Thermal conductivity examines how quickly the surface temperature of a specimen can be changed by a focused input of heat. The Presidium tester uses a hand-held wand which holds two probes. One probe applies heat and the other probe measures the temperature change. The wand is pressed onto the surface of the specimen for about five seconds. A sweeping hand shows where the thermal conductivity value for the specimen falls in relation to an overlapping series of ranges relative to known types of minerals. Each specimen was held in a non-conductive holder and analyzed at several places along the surface. In addition to confirming the mineral species of the specimens, thermal conductivity helped to ascertain whether PCZs made the thermal conductivity of any of the specimens more closely resemble the thermal conductivity of crystalline forms of silicates.

## Dichroscope

The dichroscope was used to analyze potential anisotropic differences in the paths that various frequencies of light traveled through each of the specimens (Table 1). As typical for dichroscopic analyses, each specimen was analyzed while held at different angles of view above a

non-reflective white microscopy plate. The analysis light was a white light LED flashlight [1,41].

## Polariscope

The polariscope was used to analyze aspects of the photonic control character of each specimen (Table 1). Each specimen was observed while the polarized plates were rotated to maximum and minimum occlusion. Observations occurred while each specimen was rotated 360˚. An isotropic single refractive (SR) material usually shows as brightly illuminated throughout a 360˚ revolution. An isotropic anomalous double refractive (ADR) material usually displays a dark X-like or band-shaped isogyre figure that appears and disappears or flows across the specimen every 90˚ revolution. ADR for opal requires a portion of its structure to be either crystalline or strained. A double refractive (DR) material can appear the same as an ADR material, except that a DR material will usually darken when switching between crossed and aligned polarizers [1,6,38,41–44].

# Results and discussion

## Mass and hydration

A lack of adsorption uniformity and constancy makes determining the mass, density and specific gravity of opal challenging [4,5,42]. Variable sizes and assemblages of the silicate nanospheres of opals results in great porosity diversity. These pores allow each opal specimen to adsorb/desorb $H_2O(l)$ to varying depths. Hydration levels for each opal specimen vary from 6 to 20% by volume and can be higher. Each opal specimen can adsorb/desorb a volume of water that is distinct from one another. The nanosphere character of each specimen, combined with environmental factors (e.g. external $H_2O(g)$ density and temperature), affect dehydration rates [1,4,5,38,40]. Precious opals, without $H_2O(l)$ contact, will have sub-maximum hydration due to evaporation rates and down to sub-surface levels that will vary from one specimen to another [36,37]. It is impossible, with current instrumentation, to determine the time necessary for a particular specimen to reach minimum hydration. Gemologists note more than a year of careful dehydration can be required for opals to reach stable, minimum hydration and less than a day of soaking in $H_2O(l)$ to reach maximum hydration. Amorphous silicates swell and shrink in positive correlation with internal hydration levels [43]. Once it is no longer at maximum or minimum hydration, accurate determinations of mass, density, volume and specific gravity are only transitorily valid for opal [4,5,42]. Hence, for the present research, each specimen was examined after being allowed to reach maximum hydration.

Determinations of mass, minimum/maximum hydration levels, density, volume and specific gravity (SG) were completed for each specimen (Table 2). All specimen values were within the ranges expected for precious opal [1,4,5,45,46]. The hydrated versus dehydrated differences in mass ranged from minimal for specimens 1, 2, 4 and 8 to notable for specimens 3, 5, 6 and 7, most notably for specimen 5. A comparison of the density, specific gravity and mass values for each specimen at maximum hydration show that specimens 3, 4 and 8 have the most unique combinations of these data. Specimens 3 and 5 became hazy translucent white if more than a few minutes elapsed without immersion in $H_2O(l)$, which is understandable given their maximum hydration levels (Table 2). Hydration-related changes in appearance did not appear to be significantly related to the CPP property (S1 Text General Observations). There was insufficient evidence to draw conclusions regarding CPP in relation to the data of Table 2.

**Table 2. Measured, calculated and observed, basic data.**

| Analysis (Units) | Specimen 1 | Specimen 2 | Specimen 3 | Specimen 4 | Specimen 5 | Specimen 6 | Specimen 7 | Specimen 8 |
|---|---|---|---|---|---|---|---|---|
| Material type | Precious opal | Precious opal | Precious opal | Precious opal | Precious opal | Precious opal | Precious opal | Precious opal |
| Dimensions (mm) | 8.1 x 7.4 x 3.9 | 7.9 x 5.7 x 4.8 | 10.5 x 8.6 x 3.5 | 8.1 x 6.4 x 3.7 | 14.4 x 10.5 x 3.5 | 11.9 x 8.8 x 4.4 | 10.6 x 10.1 x 8.6 | 12.4 x 9.3 x 6.8 |
| Shape | Oval cabochon | Oval cabochon | Oval cabochon | Oval cabochon | Oval cabochon | Oval cabochon | Octahedral cabochon | Oval cabochon |
| Clarity | Transparent | Transparent | Transparent | Transparent | Transparent | Transparent | Transparent | Transparent |
| Visible response, poly λ light | + play-of-color + contra luz | + play-of-color + contra luz | + play-of-color + contra luz | + play-of-color + contra luz | + play-of-color + contra luz | + play-of-color + contra luz | + play-of-color + contra luz | + play-of-color + contra luz |
| Body color, poly λ light | Colorless | Colorless | Colorless | Colorless | Colorless | Colorless | Orange | Colorless |
| Mass (g), dehydrated | 0.252 | 0.248 | 0.367 | 0.215 | 0.370 | 0.432 | 0.496 | 0.882 |
| Mass (g), hydrated | 0.259 | 0.250 | 0.419 | 0.218 | 0.482 | 0.487 | 0.540 | 0.900 |
| % $H_2O$, full hydration | 2.70 | 0.80 | 12.41 | 1.38 | 23.24 | 11.29 | 8.15 | 2.00 |
| $H_2O$ density (g/mL), 21.4˚C | 0.9978822 | 0.9978822 | 0.9978822 | 0.9978822 | 0.9978822 | 0.9978822 | 0.9978822 | 0.9978822 |
| Specific gravity | 1.0974576 | 1.0964912 | 1.0883117 | 1.1978022 | 1.0321199 | 1.0274262 | 1.0285714 | 1.0123735 |
| Density (g/mL) | 1.0997868 | 1.0988183 | 1.0906214 | 1.2003443 | 1.0343104 | 1.0296067 | 1.0307544 | 1.0145220 |
| Volume (mL) | 0.2355002 | 0.2275171 | 0.3841846 | 0.1816146 | 0.4660110 | 0.4729962 | 0.5238882 | 0.8871173 |

## CPP

Each of the specimens manifested the CPP property repeatedly (Figs 1 and 2) over a period of three years. CPP was displayed under benign conditions that did not require the addition of exogenous thermal, electronic or photonic energy. CPP specimens propagated a variety of incident light shapes and colors (Fig 3) without the CPP events being limited by PCZs or the amorphous matrices. A typical PCZ diffracted light independently of the other zones (play-of-color) [1,6]. A fully illuminated PCZ, had, essentially, an on/off photonic response, whereby it displayed play-of-color over the full PCZ. In contrast, a CPP event displayed across and flowed over multiple PCZs. Furthermore, some CPP events were color invariant and some displayed transition of color over the visible spectrum as a specimen and/or light source were rotated relative to the viewer. (S1–S8 Figs).

## Symmetry

The specimens displayed a previously undocumented photonic rotational symmetry. For a typical amorphous material, incident PPOI would match the transmitted PPOI or reflect off the bottom of the specimen to generate a specular mirror image of incident PPOI [19]. However, the research specimens rotated the incident light axially before it was reflected off of or passed through the distal side of a CPP specimen. The specimens rotated axially the incident PPOI for transmitted PPOI, reflected PPOI and the CPP property (Fig 1). Rotation of incident light by opal was not documented as having been observed prior to this research.

## Polychromatic light sources

As described, a series of five polychromatic, visible light sources was made incident upon each of the specimens (Table 1), including: compact fluorescent (CFL), LED, fiber optic polychromatic and fiber optic monochromatic. Each source had a unique colorimetric profile (Fig 4).

The application of multiple colorimetrically different light sources provided a means to conduct comparative analyses of the CPP and other photonic responses of each of the specimens. Further, each source had a different physical bulb shape (Fig 3). Thus, the detection of CPP, PPOI, rotational symmetry and other photonic properties was simplified.

### CFL

A polychromatic daylight CFL was made incident on each specimen (Table 1) (Fig 3). The specimen responses were varied (Fig 7). All specimens were transparent. Seven of the specimens displayed some level of CPP. Only specimen 4 did not display CPP. Specimen 3 displayed contra luz CPP. The other seven specimens did not display contra luz CPP. Specimens 2, 3, 7 and 8 displayed transition of color of CPP events. All specimens displayed non-CPP play-of-color. Seven specimens displayed some level of non-CPP contra luz. Only specimen 5 displayed no non-CPP contra luz. Although the incident light was white, the reflected PPOI was white for specimens 1, 3 and 8, yellow for specimens 4 and 6, pinkish yellow for specimens 2 and 3 and orange for specimen 7 (S1 Text Compact Fluorescent Light).

### LED

A 22 cm long lamp with three rows of eleven polychromatic LED bulbs was made incident on each specimen (Table 1). Specimen responses were varied (Fig 8). The pattern of the LED

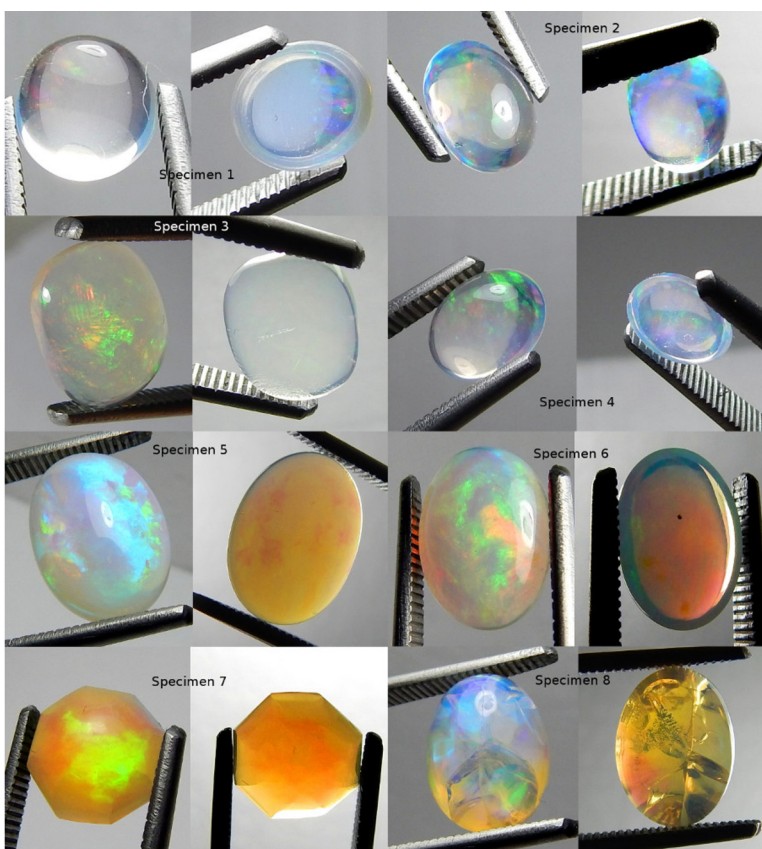

**Fig 7. Photographs of all specimens under CFL polychromatic light.** Two photos per specimen, one each showing play-of-color (left) and contra luz (right).

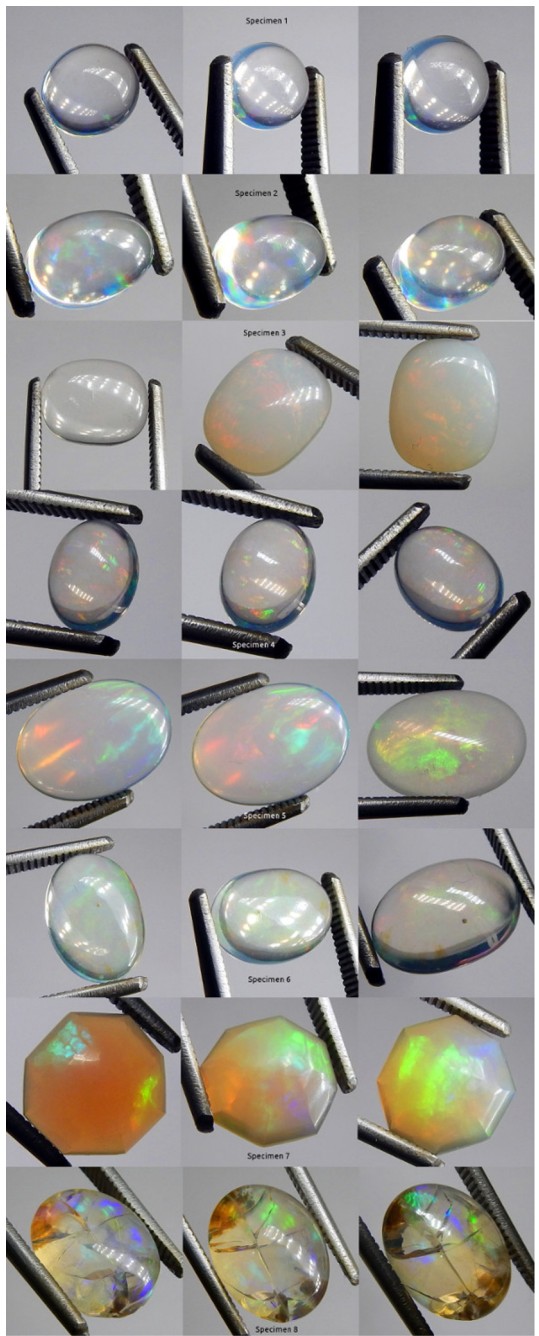

**Fig 8. Photographs of all specimens under LED polychromatic light.** Three photos per specimen.

bulbs made the rotation of the incident light more readily identifiable (Fig 3). All specimens were transparent. Seven of the specimens displayed some level of CPP. Only specimen 3 did not display CPP. Specimens 2, 3 and 4 displayed contra luz CPP. Specimens 1, 5, 6, 7 and 8 did not display contra luz CPP. Specimens 2, 3, 4, 6, 7 and 8 displayed transition of color of CPP events. Only specimens 1 and 5 did not display transition of color of CPP events. All specimens displayed non-CPP play-of-color. Specimens 1, 2, 3, 4 and 6 displayed some level of non-CPP contra luz. Specimens 5, 7 and 8 did not display non-CPP contra luz. Although the incident

light was white, the reflected PPOI was white for specimens 1, 7 and 8, yellow for specimens 3, 4 and 6, pinkish yellow for specimen 2 and no response for specimen 5 (S1 Text LED).

## Fiber optic

Two different clusters of fiber optic cables, illuminated with a daylight halogen light source, were made incident on each specimen (Table 1). One observation was conducted with a set of cables that were not dyed and another observation was conducted with a set that was dyed quadchromatically. The patterns and/or colors of the cables made the CPP property more readily identifiable (Fig 3).

The first type of cluster was undyed polychromatic fiber optic cables. Specimen responses were varied (Fig 9). All specimens were transparent. Specimens 6 and 8 had mild haze. Seven specimens displayed some level of CPP. Only specimen 3 did not display CPP. Specimens 2 and 4 displayed contra luz CPP. Specimens 1, 3, 5, 6, 7 and 8 did not display contra luz CPP. Specimens 2, 3, 4, 6 and 7 displayed transition of color of CPP events. Only specimens 1, 5 and 8 did not display transition of color of CPP events. All specimens displayed non-CPP play-of-color. Specimens 1, 2, 3 and 4 displayed some level of non-CPP contra luz. Specimens 5, 6, 7 and 8 did not display non-CPP contra luz. Although the incident light was white, specimens 1, 3, 4 and 7 displayed white reflected PPOI, specimens 2, 5 and 6 displayed red reflected PPOI and specimen 8 displayed green reflected PPOI (S1 Text Fiber Optic Cluster, Polychromatic).

The second type of cluster was dyed quadchromatically via the combination of four mono-chromatic clusters of three fiber optic cables (Table 1) (Fig 10). Each cluster was dyed one of four colors (red, yellow, green and blue). Applying quadchromatic light was a means to exam-ine the photonic responses of each specimen to a combination of the four monochromatic fiber optic wavelengths. All specimens were transparent. Seven specimens displayed some level of CPP. Only specimen 1 did not display CPP. Specimens 2 and 4 displayed contra luz CPP. Specimens 1, 3, 5, 6, 7 and 8 did not display contra luz CPP. Specimens 2, 4, 5, 6, 7 and 8 showed positive CPP color transformations. Only specimens 1 and 3 showed negative CPP color transformations. Specimens 1, 2, 3, 5, 6 and 7 displayed non-CPP play-of-color. Speci-mens 4 and 8 did not display non-CPP play-of-color. Specimens 1 and 2 displayed some level of non-CPP contra luz. Specimens 3, 4, 5, 6, 7 and 8 did not display non-CPP contra luz. The reflected PPOI matched all four incident PPOI colors for each of the specimens (S1 Text Fiber Optic Cluster, Quadchromatic).

## Monochromatic light sources

In visible light, a series of monochromatic light sources were made incident on each specimen, including: four fiber optic clusters, five lasers and three UV sources (Tables 1, 3 and 4). Differ-ences in incident chromaticity affected the presence, strength and character of each of the observed photonic properties for each specimen. Applying light from clusters of monochro-matic and polychromatic fiber optics provided a means to conduct comparative analyses of the photonic responses of the specimens (Fig 3) (S1 Text Fiber Optic Cluster, Monochromatic). Applying laser light provided a means to examine the photonic responses of the specimens under uni-directional mono-frequency photonic sources. Applying UV light provided a means to examine the visible light photonic responses of the specimens under non-visible pho-tonic sources.

## Fiber optic

Under the monochromatic fiber optic sources, the photonic responses changed as the light source was changed between blue, green, yellow and red. The specimens displayed a range of

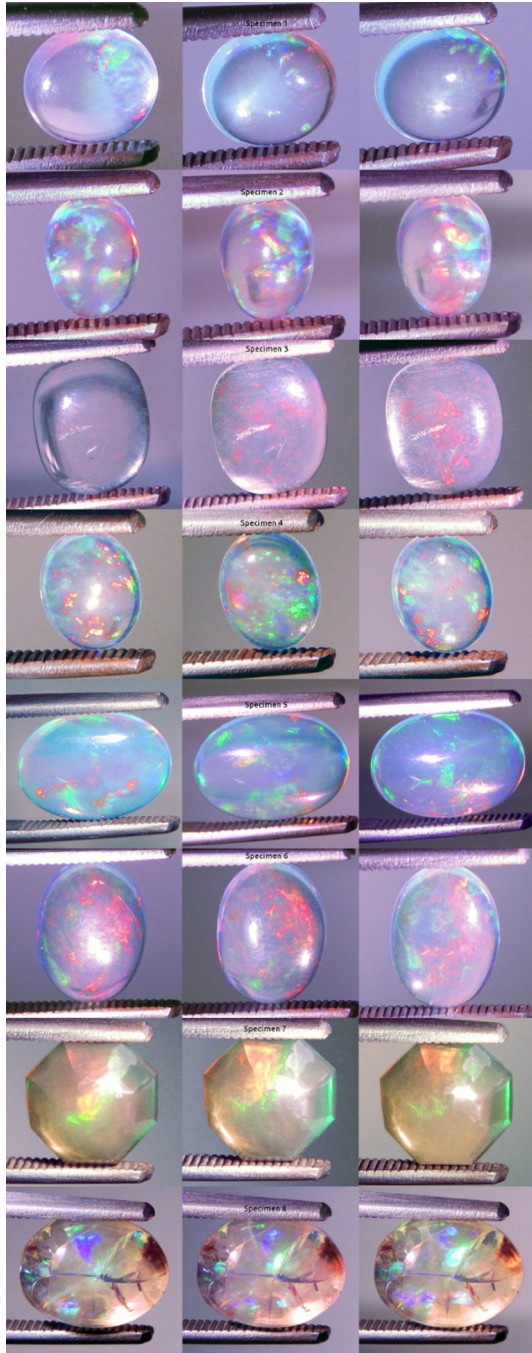

**Fig 9. Photographs of all specimens under white fiber optic polychromatic light.** Three photos per specimen.

photonic properties, including upconversion, downconversion, play-of-color, contra luz, CPP, and contra luz CPP.

Blue fiber optic light was made incident on each specimen (Fig 11). Each specimen was positive for CPP at varying levels. Specimens 1, 4 and 6 were negative for non-CPP play-of-color. Specimens 2, 3, 5, 7 and 8 were strongly positive for non-CPP play-of-color. CPP events

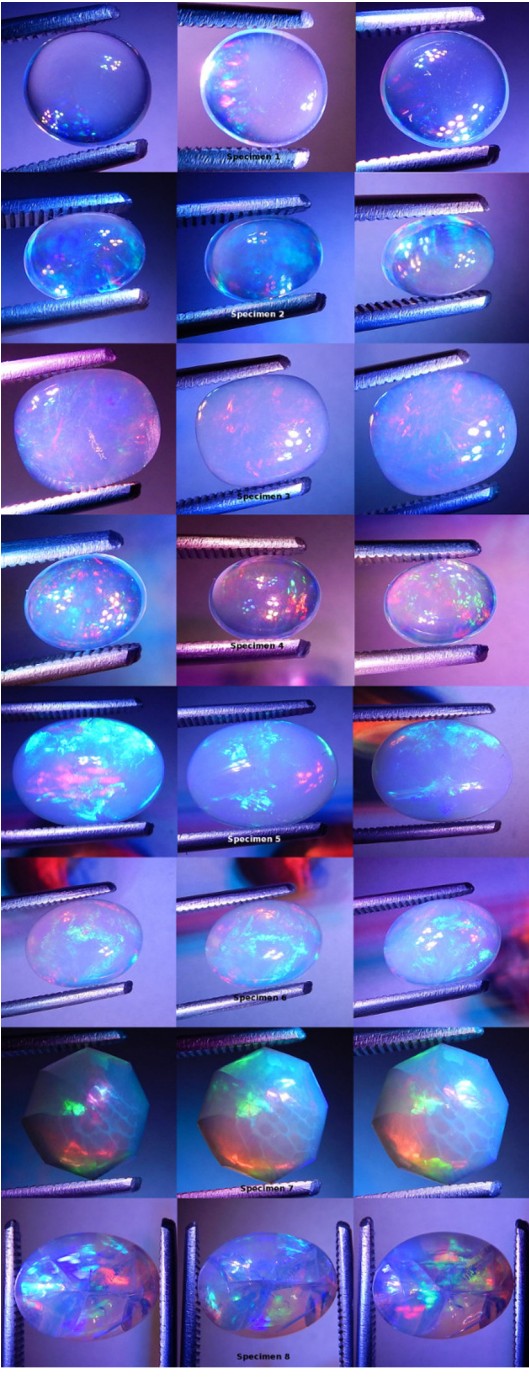

**Fig 10. Photographs of all specimens under quadchromatic fiber optic polychromatic light.** Three photos per specimen.

included from one to three colors, with mild to strong color saturation, with tiny to moderate spread and with varying amounts of blur.

Green fiber optic light was made incident on each specimen (Fig 12). Each specimen was positive for CPP at varying levels. Only specimen 1 was negative for non-CPP play-of-color. Specimens 2, 3, 4, 5, 6, 7 and 8 were varyingly positive for non-CPP play-of-color. CPP events

**Table 3. Summary of laser photonic responses.**

| Laser Analysis | Specimen 1 | Specimen 2 | Specimen 3 | Specimen 4 | Specimen 5 | Specimen 6 | Specimen 7 | Specimen 8 |
|---|---|---|---|---|---|---|---|---|
| 650 nm | + visible path<br>- transmission<br>+ smooth glow | - visible path<br>- transmission<br>+ grainy glow | + visible path<br>+ transmission<br>+ smooth glow | + partial visible path<br>- transmission<br>+ smooth glow | + visible path<br>- transmission<br>+ smooth glow | + visible path<br>- transmission<br>+ smooth glow | + visible path<br>- transmission<br>+ grainy glow | + visible path<br>- transmission<br>+ grainy glow |
| 593.5 nm | + visible path<br>+ transmission<br>+ grainy glow | - visible path<br>- transmission<br>+ grainy glow | + visible path<br>+ transmission<br>+ grainy glow | + visible path<br>- transmission<br>+ grainy glow | + visible path<br>- transmission<br>+ grainy glow | + visible path<br>- transmission<br>+ grainy glow | + visible path<br>+ transmission<br>+ grainy glow | + visible path<br>- transmission<br>+ grainy glow |
| 532 nm | + visible path<br>- transmission<br>+ smooth glow | + visible path<br>- transmission<br>+ grainy glow | + visible path<br>+ transmission<br>+ smooth glow | + visible path<br>+ transmission<br>+ smooth glow | - visible path<br>- transmission<br>+ smooth glow | + visible path<br>- transmission<br>+ smooth glow | + visible path<br>- transmission<br>+ grainy glow | + visible path<br>- transmission<br>+ grainy glow |
| 450 nm | + visible path<br>+ transmission<br>+ smooth glow | + visible path<br>+ transmission<br>+ smooth glow<br>4-ray star | + visible path<br>- transmission<br>+ smooth glow | + visible path<br>+ transmission<br>- smooth glow | - visible path<br>- transmission<br>+ smooth glow | - visible path<br>- transmission<br>+ smooth glow | - visible path<br>- transmission<br>+ smooth glow | - visible path<br>- transmission<br>+ smooth glow |
| 405 nm | + visible path<br>+ transmission<br>+ smooth glow | + visible path<br>+ transmission<br>+ smooth glow | + visible path<br>+ transmission<br>+ smooth glow | + visible path<br>+ transmission<br>+ smooth glow | - visible path<br>- transmission<br>+ smooth glow | - visible path<br>- transmission<br>+ smooth glow | - visible path<br>- transmission<br>+ smooth glow | + visible path<br>- transmission<br>+ smooth glow |

included from one to four colors, with mild to strong color saturation, with tiny to moderate spread and with varying amounts of blur.

Yellow fiber optic light was made incident on each specimen (Fig 13). Each specimen was positive for CPP at varying levels. Only specimen 4 was only very slightly positive for non-CPP play-of-color. Seven specimens were strongly positive for non-CPP play-of-color. CPP events included from three to four colors, with mild to strong color saturation, with very small to moderate spread and with varying amounts of blur.

Red fiber optic light was made incident on each specimen (Fig 14). Seven specimens were positive for CPP at varying levels. Only specimen 3 was negative for CPP. Only specimens 3 and 7 were strongly positive for non-CPP play-of-color. Specimen 7 showed only a few CPP events, but those events included the widest range of CPP colors. CPP events included from one to four colors, with mild to strong color saturation, with minimal to very strong spread and with varying amounts of blur.

## Laser

The laser-specimen interactions did not trigger the CPP property. The present research examined three aspects of laser-specimen interactions (Fig 5). First, path of traversing through. Each specimen was observed to examine each incident laser light as it traversed through and across the interior of each specimen. Second, passing through intact. Each specimen was observed to examine the extent to which each laser light passed coherently through and out the body of each specimen. This observation was made by watching for light on a white board

**Table 4. Summary of UV photonic responses.**

| UV Analysis | Specimen 1 | Specimen 2 | Specimen 3 | Specimen 4 | Specimen 5 | Specimen 6 | Specimen 7 | Specimen 8 |
|---|---|---|---|---|---|---|---|---|
| 375 nm | Smooth purple | Smooth purple | Smooth purple | Smooth purple, black, gray, white | Yellow, red & green play-of-color | Smooth purple, pink, black | Yellow, red & green play-of-color | Smooth purple, green, blue |
| 307 nm | Colorless | Colorless | Smooth purple, orange | Smooth orange, colorless | Smooth orange | Smooth purple, orange | Smooth orange | Smooth orange |
| 254 nm | Black | Greenish black | Greenish black | Greenish black | Greenish black | Black, gray, white | Greenish black | Smooth blackish violet, white |

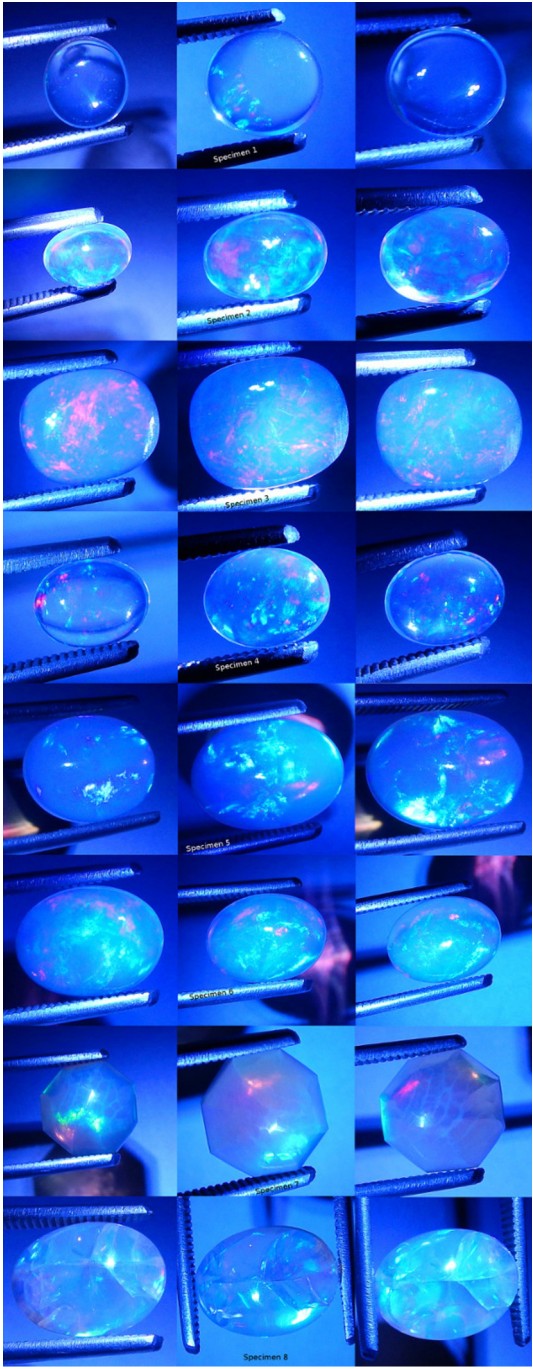

**Fig 11. Photographs of all specimens under blue fiber optic monochromatic light.** Three photos per specimen.

set up approximately 12 cm away from the side of each specimen that was distal from the incident PPOI. Third, body glow. Each specimen was observed for the presence, brightness, color and texture of body glow due to each incident laser light.

A 650 nm laser (Table 3) displayed visible, coherent paths through specimens 1, 3, 4, 5, 6, 7 and 8. The laser did not display a visible path while traversing specimen 2. The coherent path for specimen 4 was only present on the side proximal to the incident PPOI. The path for

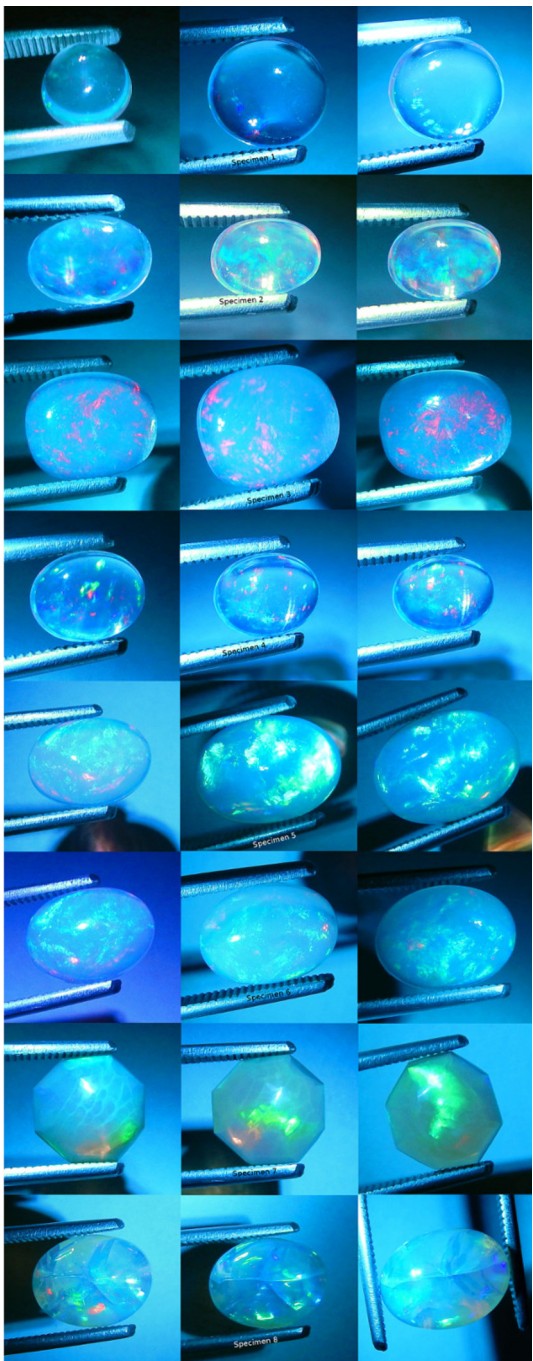

**Fig 12. Photographs of all specimens under green fiber optic mono-chromatic light.** Three photos per specimen.

specimen 5 was particularly strong. Specimen 6 displayed a wedge shaped path that was broader at incident PPOI and narrower towards the distal side. Each specimen, in which the traversing laser path was visible, displayed a path that was somewhat wider than the incident PPOI column. Second, only specimen 3 transmitted coherent laser light out of the distal side. Even so, it passed only a very small, weak dot, relative to the incident laser light column. All other specimens experienced total internal reflection, such that each specimen blocked

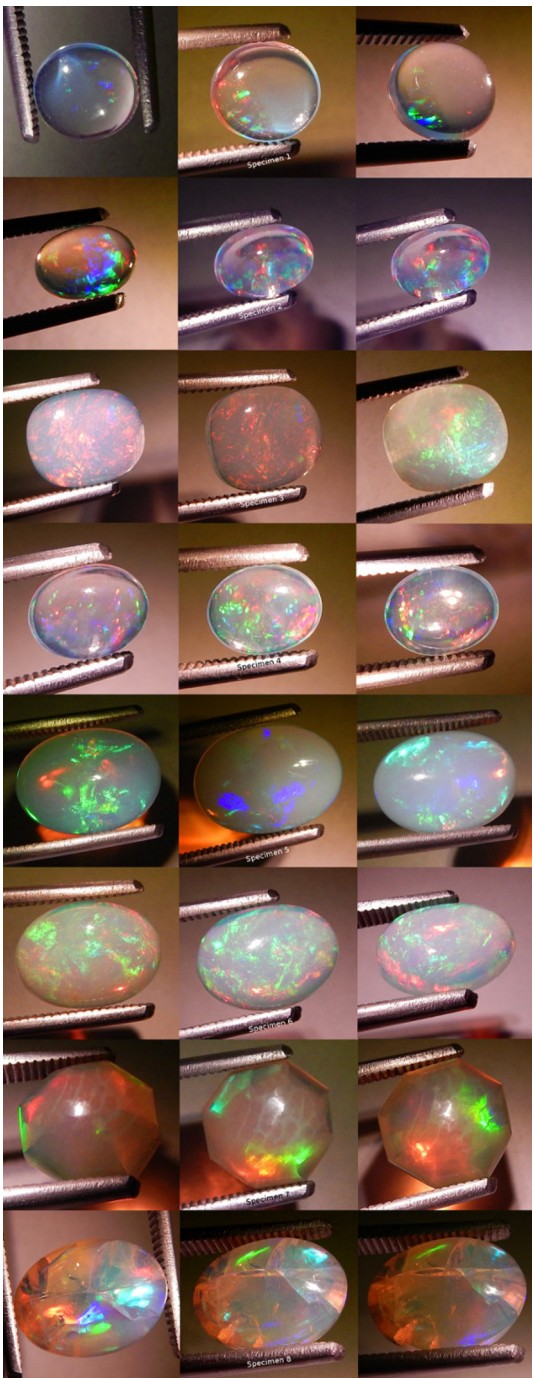

**Fig 13. Photographs of all specimens under yellow fiber optic monochromatic light.** Three photos per specimen.

coherent laser light from transmitting through the distal side. Third, all of the specimens displayed body glow. The laser caused a smooth body glow of portions of specimens 1, 3, 4, 5 and 6. The non-glowing areas completely absorbed the incident laser and appeared black. Specimen 4 displayed a particularly strong smooth glow. The laser caused a grainy body glow of specimens 2, 7 and 8. Specimens 2 and 7 had strong grainy glows, whereas specimen 8 had only a mild grainy glow. Specimen 2 exhibited the most unique responses to the 650 nm laser.

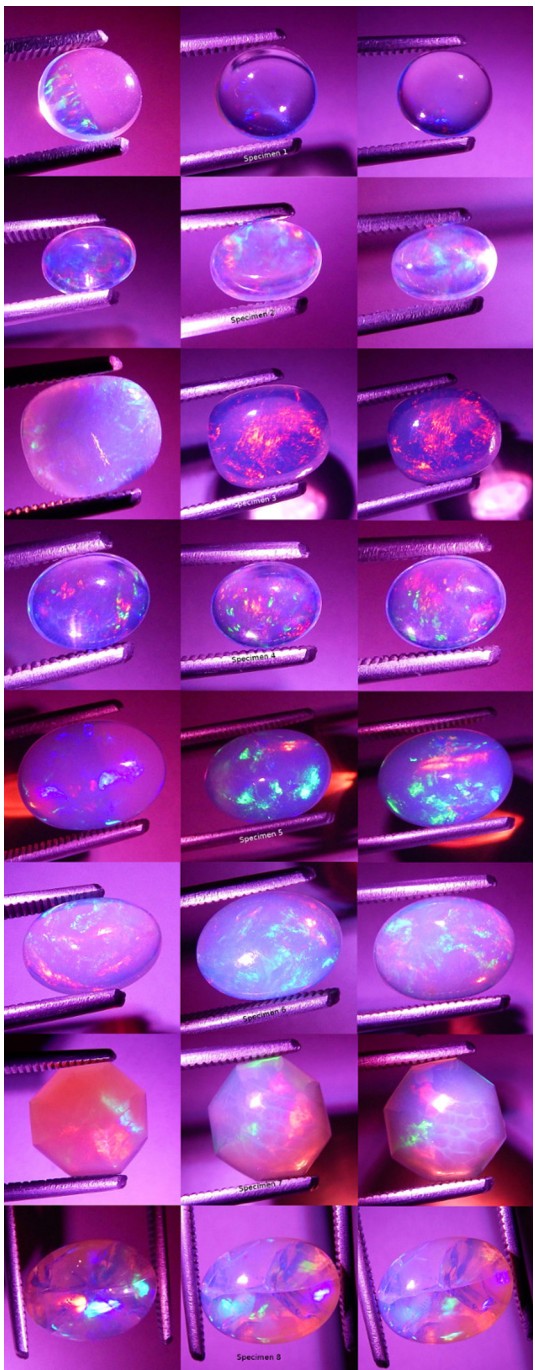

**Fig 14. Photographs of all specimens under red fiber optic monochromatic light.** Three photos per specimen.

A 593.5 nm laser (Table 3) displayed visible, coherent paths through specimens 1, 3, 4, 5, 6, 7 and 8. The paths for specimens 1, 4 and 8 were of mild strength. The path for specimen 5 was particularly strong, but was wider than the paths of the other specimens. The path for specimen 6 showed diminishing strength towards the distal side. The laser did not display a visible path while traversing specimen 2. Each specimen, in which the traversing laser path was visible, displayed a path that was somewhat wider than the incident PPOI column. Second,

specimens 1, 3 and 7 transmitted coherent laser light out of the distal side. Even so, the lasing dot for specimen 3 was diffuse and the lasing dots for specimens 1 and 7 were very small and weak, relative to the incident laser light column. Specimens 2, 4, 5, 6 and 8 experienced total internal reflection, such that each specimen blocked coherent laser light from transmitting through the distal side. Third, the laser caused all specimens to display strongly grainy body glow textures. Specimens 2 and 6 exhibited the most unique responses to the 593.5 nm laser.

A 532 nm laser (Table 3) displayed visible, coherent paths through specimens 1, 2, 3, 4, 6, 7 and 8. The path through specimen 2 was of mild strength. Specimen 6 displayed a wedge shaped path that became narrower towards the distal side. The laser did not display a visible path while traversing specimen 5. Each specimen, in which the traversing laser path was visible, displayed a path that was somewhat wider than the incident PPOI column. Second, specimens 3 and 4 transmitted coherent laser light out of the distal side. The exit dot for specimen 3 was diffuse and the exit dot for specimen 4 was very small, relative to the incident laser light column. Specimens 1, 2, 5, 6, 7 and 8 experienced total internal reflection, such that each specimen blocked coherent laser light from transmitting through the distal side. Third, the laser caused a smooth body glow of specimens 1, 3, 4, 5 and 6. Specimen 5 displayed smooth glow over only one third of the specimen on the incident side and was black over two thirds of the distal side. The laser caused specimens 2, 7 and 8 to display a grainy glow. Specimen 7 displayed only mildly grainy texture. Of note, specimen 6 reflected, particularly strongly, the incident PPOI. Specimens 5 and 6 exhibited the most unique responses to the 532 nm laser.

A 450 nm laser (Table 3) displayed visible, coherent paths through specimens 1, 2, 3 and 4. The paths through specimens 1, 2 and 4 were of mild strength and were especially mild for specimens 2 and 4. The laser did not display a visible path while traversing specimens 5, 6, 7 and 8. Each specimen, in which the traversing laser path was visible, displayed a path that was at least slightly wider than the incident PPOI column. Second, specimens 1, 2 and 4 transmitted coherent laser light out of the distal side. The exit dots for specimens 2 and 4 were very small, relative to the incident laser light column. Specimens 3, 5, 6, 7 and 8 experienced total internal reflection, such that each specimen blocked coherent laser light from transmitting through the distal side. Third, the laser caused a smooth body glow of specimens 1, 2, 3, 5, 6, 7 and 8. Specimen 1 displayed only a very mild smooth glow. Specimens 5, 6, 7 and 8 displayed smooth glow over only one third of the specimen on the incident side and black over two thirds of the distal side. The laser was almost completely absorbed by specimen 4, such that the specimen was almost entirely black with only a very slight smooth purple glow. Of note, this laser caused specimen 2 to display a four-ray star, which was previously unknown to occur in opals in response to lasers. Specimens 5, 6, 7 and 8 appeared translucent in response to this laser. Specimens 2 and 4 exhibited the most unique responses to the 450 nm laser.

A 405 nm laser (Table 3) displayed visible, coherent paths through specimens 1, 2, 3, 4 and 8. Yet, the paths through specimens 1, 2 and 8 were barely visible. The laser did not display a visible path while traversing specimens 5, 6 and 7. Each specimen, in which the traversing laser path was visible, displayed a path that was somewhat wider than the incident PPOI column. Second, specimens 1, 2, 3 and 4 transmitted coherent laser light out of the distal side. The exit dot for specimen 1 was very small, relative to the incident laser light column. Specimens 5, 6, 7 and 8 experienced total internal reflection, such that each specimen blocked coherent laser light from transmitting through the distal side. Third, the laser caused a smooth body glow of all specimens. Specimens 5 and 6 displayed smooth purple glow over only one third of the specimen on the incident side and was slightly purplish black over two thirds of the distal side. Specimen 7 exhibited partial total absorption with black over one third of the specimen on the incident side and downconversion to green over two thirds of the distal side. Specimen 8 exhibited downconversion by displaying a smooth green dot at incident PPOI and a smooth

purple body glow over its remainder. While specimens 5 and 6 were interesting, specimens 7 and 8 exhibited the most unique responses to the 405 nm laser.

### UV

Three mono-frequency UV sources were made incident on each specimen (Table 1). UV-specimen interactions were observed to research the CPP property as well as other photonic properties of the specimens. These observations were made while freely changing the relative positions of the viewer, specimens and light sources. None of the specimens displayed CPP or reflected PPOI in response to UV light. Even so, play-of-color was observed in response to UV light, which had not been documented previously.

UV light of 375 nm (Table 4) caused specimens 1, 2, 3 and 8 to exhibit smooth purple violet colors that ranged in strength from mildest for specimen 1 to strongest for specimen 3. Specimens 4 and 6 exhibited black that faded, distally, into gray and white with a smooth purple violet zone. All specimens exhibited Stokes downconversion. Specimens 5 and 7 exhibited downconversion via play-of-color, showing zones of multiple colors that included yellow, green and red. Specimen 8 exhibited downconversion via central smooth purple violet with an outer rim of blue and green. Specimen 6 exhibited black and downconversion via smooth pink and purple. The photonic responses of specimens 5 and 7 with 375 nm UV had not been reported previously. Specimens 5, 7 and 8 exhibited the most unique responses.

UV light of 307 nm (Table 4) caused specimens 1 and 2 to become colorless and transparent. Specimens 3 and 6 exhibited smooth purplish orange. Specimen 4 became mostly colorless with one thin zone of smooth orange. Specimens 5, 7 and 8 exhibited smooth orange. Specimen 7 displayed very deep orange. Specimens 1, 2 and 4 exhibited the most unique responses.

UV light of 254 nm (Table 4) caused specimens 2, 3, 4, 5 and 7 to exhibit smooth greenish black. Specimens 1 exhibited only black. Specimen 8 exhibited smooth blackish purple violet with a zone of white on one end. Specimen 6 exhibited black that faded into gray and then white. Specimen 1 displayed the most unique responses.

### Spectrophotometry

Spectrophotometric scans, over four orientations (Fig 6), were conducted on each specimen (Tables 1 and 5) to research the nature of CPP and to test whether the photonic absorption properties of each specimen were internally uniform or directionally sensitive. Each specimen

**Table 5. Comparison summary of select instrumentation analyses.**

| Analysis Method | Specimen 1 | Specimen 2 | Specimen 3 | Specimen 4 | Specimen 5 | Specimen 6 | Specimen 7 | Specimen 8 |
|---|---|---|---|---|---|---|---|---|
| Spectrophotometer | Typical | Some variations, especially at higher λ | Not typical at any λ | Some variations, especially at lower and higher λ | Not typical at any λ | Scans 1 and 2 were not typical at any λ | Not typical at any λ | Not typical at any λ |
| Refractive Index | 1.420 | 1.380 | 1.395 | 1.380 | 1.432 | 1.459 | 1.362 | 1.365 |
| Thermal Conductivity | Typical | Typical | Typical | Typical | Typical | Typical | Typical | Typical |
| Dichroscope | Monochroic colorless body | Monochroic colorless body, Dichroic contra luz | Monochroic colorless body, Dichroic contra luz | Monochroic colorless body, Dichroic contra luz | Monochroic colorless body, Dichroic contra luz | Monochroic colorless body, Dichroic contra luz | Monochroic colorless body, Dichroic contra luz | Monochroic colorless body, Dichroic contra luz |
| Polariscope | Isotropic singly refractive ADR | Isotropic singly refractive ADR | Isotropic singly refractive ADR | Isotropic singly refractive ADR | Isotropic singly refractive | Isotropic singly refractive ADR | Isotropic singly refractive | Isotropic singly refractive |

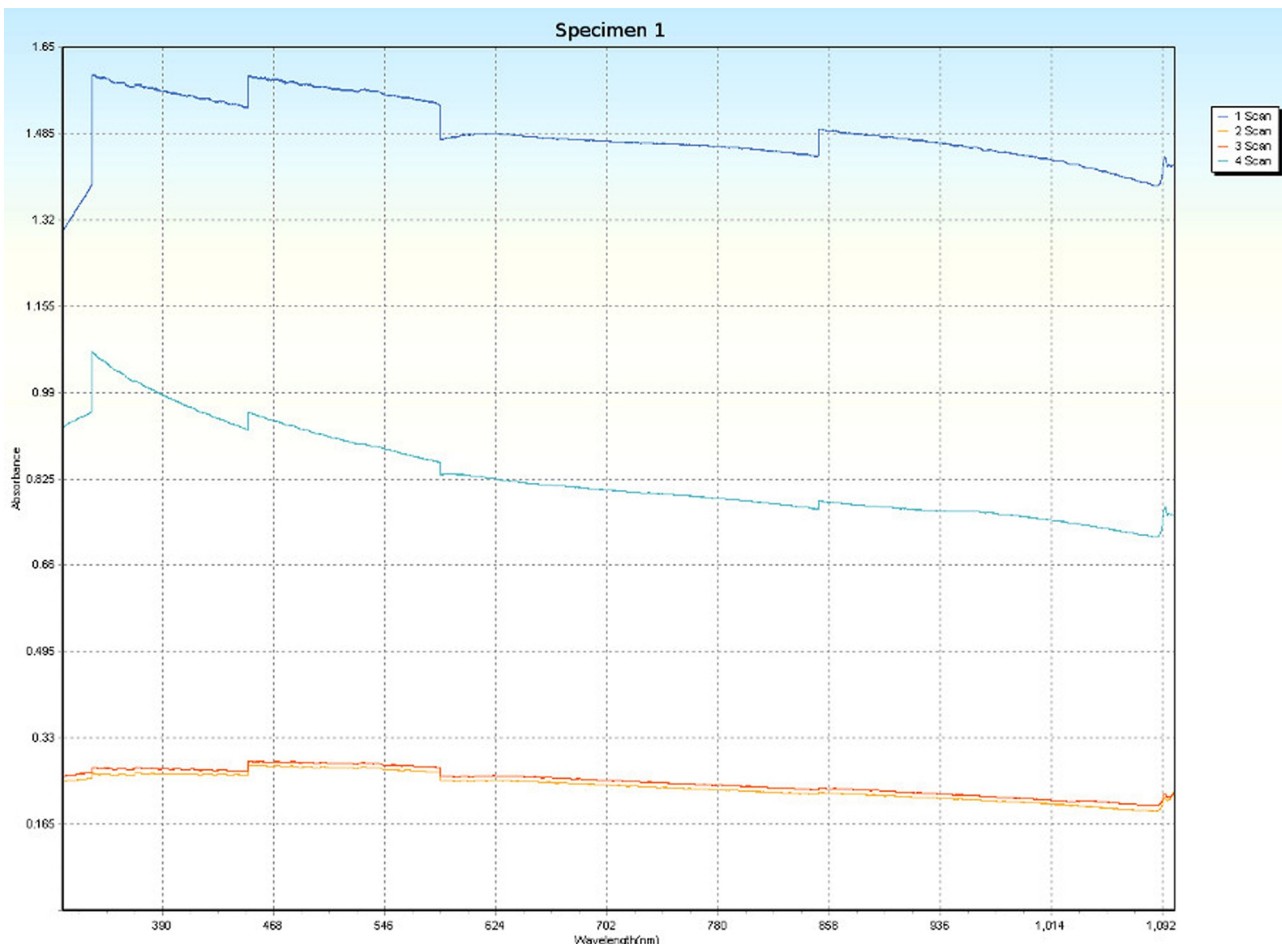

**Fig 15. Spectrophotometric scans of specimen 1.** a. top to bottom, b. bottom to top, c. side-to-side narrow, d. side-to-side long.

was a non-crystalline silicate with PCZs and visibly uniform color. Hence, each specimen was expected to have an overall non-uniform distribution of molecular structure that did not affect photonic flow. Therefore, it was tested whether properties of these specimens spectrophotometrically interfered with photonic absorption with orientation sensitivity. For most of the specimens, the solid-state spectrophotometric scans showed significant analytical differences from scan to scan (Figs 15–22). Although each scan showed a generally negative correlation with absorption wavelength, some of the scans were very complex. The spectrophotometric scans showed that varying wavelengths of light were absorbed by specimens that should have been almost entirely photonically non-absorptive over visible light because the specimens were either colorless, hazy white or orange. A comparison of the four scans of each specimen revealed that the photonic absorption by these uniformly colored amorphous silicate materials could be affected by specimen orientation. (S1 Text Spectrophotometer).

## Refractive index

Refractive index (RI) analysis was conducted on each specimen. RI identified each specimen as non-crystalline opal silicate [1,5,6,38,39] (Tables 1 and 5). Thus, the total volume, alignment and dispersion of PCZs in these precious opal specimens were not sufficient to cause a significant change in RI from that of amorphous opal.

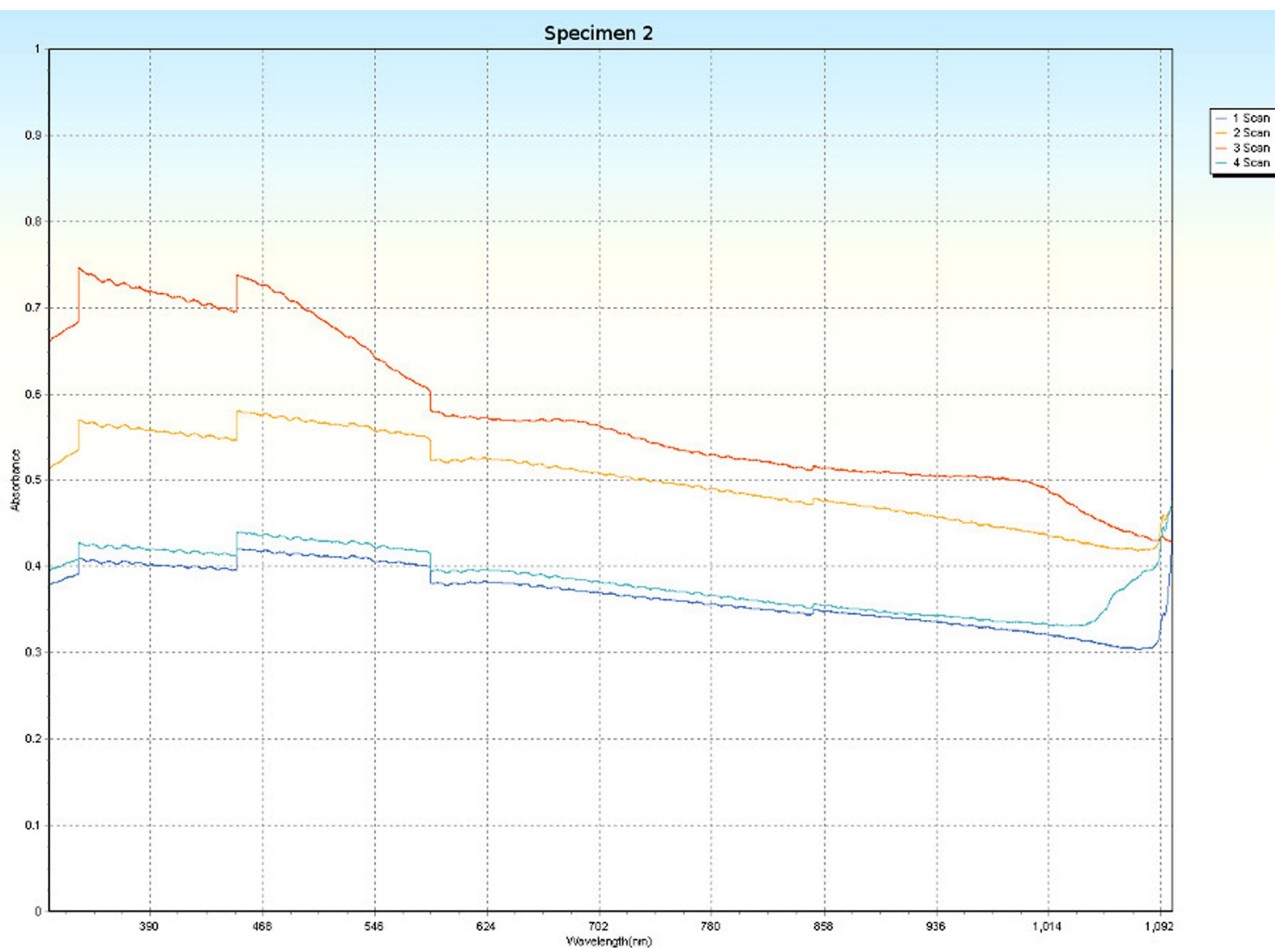

**Fig 16. Spectrophotometric scans of specimen 2.** a. top to bottom, b. bottom to top, c. side-to-side narrow, d. side-to-side long.

### Thermal conductivity

Thermal conductivity analysis identified each specimen as a non-crystalline silicate [1,4,38,40] (Tables 1 and 5). Even so, there were slight differences in readings between each specimen. Thus indicating that the total volume, alignment and dispersion of PCZs in these precious opal specimens resulted in some minor variability but were not sufficient to cause a significant change in thermal conductivity from that of amorphous opal.

### Dichroscope

A dichroscopic analysis was conducted on each specimen (Tables 1 and 5). This test showed that each specimen was, as expected, synchromatic [1,4,41]. Surprisingly, seven of the specimens displayed contra luz when viewed through the dichroscope. Only specimen 1 had no dichroscopic contra luz response. When using the dichroscope to view each specimen in areas actively displaying contra luz, asynchromatic dichroism was observed. Dichroscopic observations of the contra luz property had not been documented previously.

### Polariscope

A polariscopic analysis was conducted on each specimen (Tables 1 and 5). As expected, the polariscopic examinations showed that each of the specimens was isotropic. Specimens 5, 7

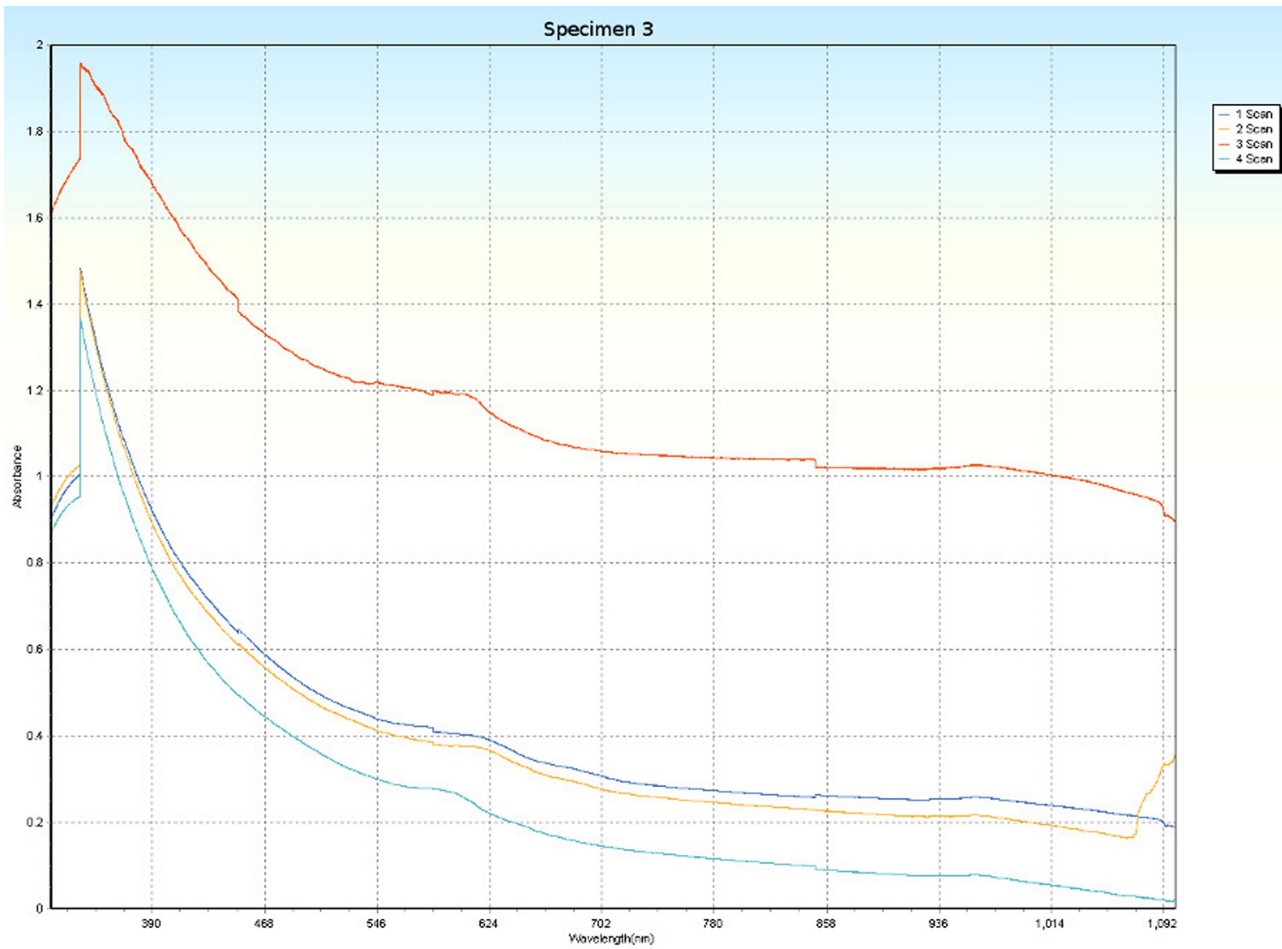

**Fig 17. Spectrophotometric scans of specimen 3.** a. top to bottom, b. bottom to top, c. side-to-side narrow, d. side-to-side long.

and 8 were SR. Specimens 1, 2, 3, 4 and 6 were ADR. As expected for each ADR specimen, the orientation of the X-like isogyre figure did not change as each specimen was rotated [42]. ADR was confirmed because no specimens darkened when switching between crossed and aligned polarizers. SR/ADR was confirmed because significant crystallinity was not supported via RI or thermal conductivity (S1 Text Polariscope).

## General

Specimen 1 was the only specimen in which only a portion of the specimen (about half) was precious contra luz opal and the other portion was common opal (no play-of-color and no contra luz). CPP events appeared in the precious opal portion. CPP events did not appear in the common opal portion. Reflected and transmitted PPOI appeared in the common opal portion. Hence, CPP seems to be a property related to ordered microspheres.

Specimens 1 and 4 were the only specimens that displayed asterism. This property manifested under monochromatic fiber optic light. Asterism is a rare property for natural opal that had not been previously documented to occur under monochromatic light.

Specimen 2 displayed a particularly unusual CPP event. This event may have been related to having the strongest CPP and non-CPP contra luz responses under all light sources. In particular, under green fiber optic light, a significant portion of specimen 2 responded as a unified

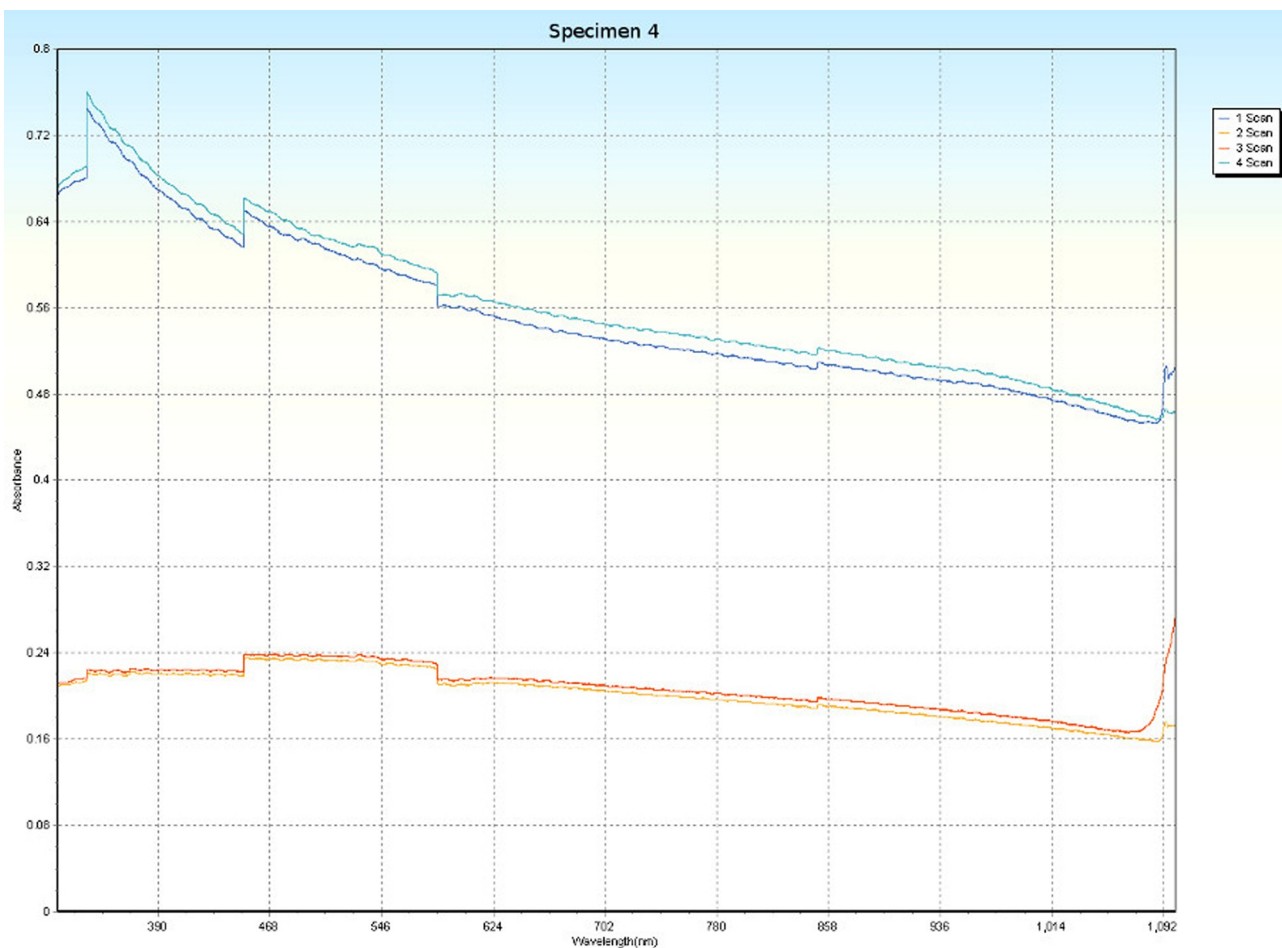

**Fig 18. Spectrophotometric scans of specimen 4.** a. top to bottom, b. bottom to top, c. side-to-side narrow, d. side-to-side long.

prismatic contra luz, with many CPP events, displaying red closest to the incident PPOI and then progressing through orange, yellow, green, blue and finally violet was most distal from the incident PPOI. This CPP contra luz response contradicts the common thought that precious opal is comprised of randomly oriented islands of PCZs that lack the ability to exert photonic control over a large, macroscopic volume [1,3–6].

Specimen 3 had unique property changes that resulted from it being especially sensitive to changes in internal $H_2O$ levels. While starting out as transparent, specimen 3 started to become hazy translucent white after only about five minutes of air exposure (e.g. removed from $H_2O$(l) immersion). This haze response was similar to that demonstrated by specimen 5. Non-CPP play-of-color was minimal at full hydration and became moderately strong after a few minutes of air exposure. Also, the increased haze caused CPP and non-CPP play-of-color events to be more readily visible due to increased contrast.

For specimen 3, the colors of non-CPP play-of-color were highly dependent on the angle of incidence. Regardless of the light source, mostly reds/oranges displayed when the angle of incidence approached the perpendicular relative to the viewer, which changed to mostly greens/blues when the angle of incidence approached the horizontal relative to the viewer (S9). Furthermore, the proximal side of this specimen was significantly flatter than those of the other

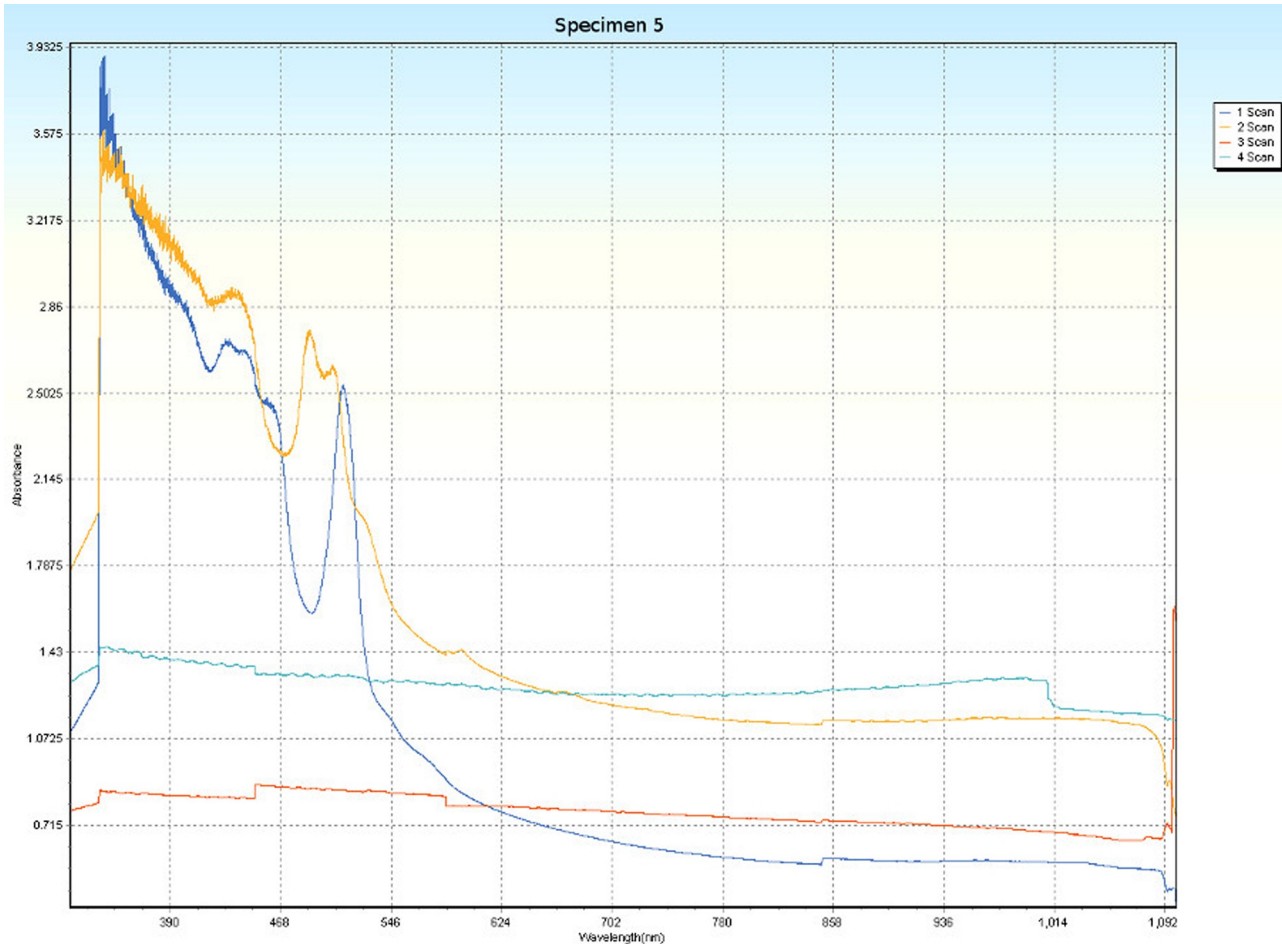

**Fig 19. Spectrophotometric scans of specimen 5.** a. top to bottom, b. bottom to top, c. side-to-side narrow, d. side-to-side long.

specimens. Hence, the reflected PPOI propagated had a wider angle, which interfered with the visibility of the reflected PPOI from the incident side.

Specimens 4, 5 and 6 had an unusual haze response to incident light. Each of these specimens formed a mild haze proximal to the incident PPOI for incident monochromatic fiber optic light sources. In contrast, other specimens that formed haze and other haze-inducing light sources on specimens 4, 5 and 6, formed it proximal to the reflected PPOI.

Specimen 5 demonstrated sensitivity to changes in internal $H_2O$ levels. This specimen was completely transparent when fully hydrated and started to become hazy translucent white after about four minutes of air exposure. This haze response was similar to that demonstrated by specimen 3. Even so, hydration-related changes in transparency did not appear to affect CPP or non-CPP play-of-color events for this specimen.

Specimens 5, 6 and 7 demonstrated unusual reflected PPOI events. Reflected PPOI was red orange, not the white reflected PPOI displayed by most of the other specimens. These unusual events were observed under each of the four monochromatic fiber optic sources.

Specimen 6 was unique among the test specimens because of the visible indications that resulted from it being somewhat sensitive to changes in internal $H_2O$ levels. While there was no haze response, after a few days of exposure to air, specimen 6 started to respond differently to incident light. For example, under the quad-cluster fiber optic incident light, the specimen

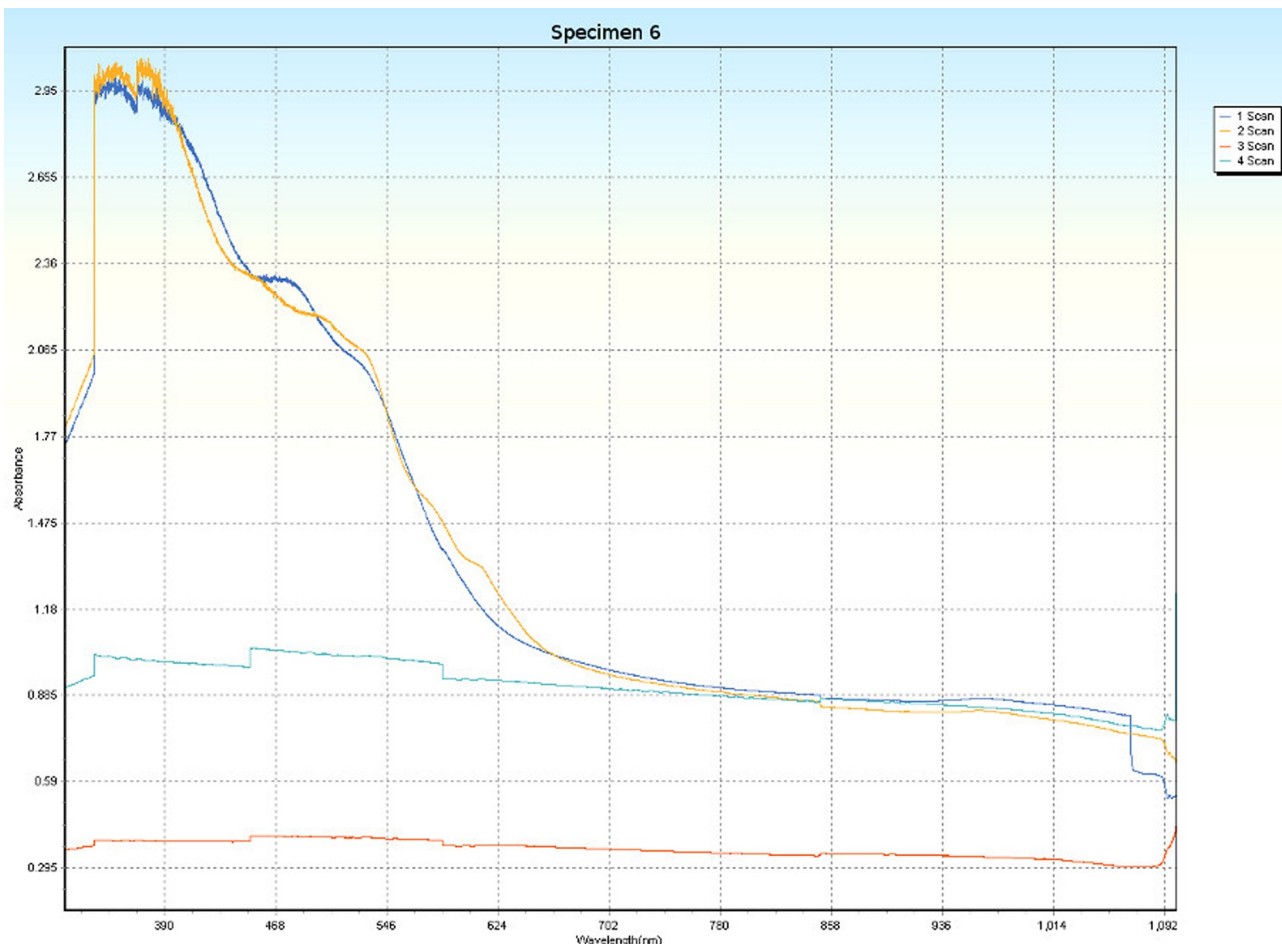

**Fig 20. Spectrophotometric scans of specimen 6.** a. top to bottom, b. bottom to top, c. side-to-side narrow, d. side-to-side long.

did not propagate all four incident wavelengths for transmitted or reflected PPOI. Yet, hydration-related changes did not appear to affect CPP events.

Specimen 7 had property changes that resulted from it being somewhat sensitive to changes in internal $H_2O$ levels. While there was no haze response, after many days of exposure to air, specimen 7 started to respond differently to incident light. For example, under the quad-cluster fiber optics incident light, the specimen did not always propagate incident blue. Instead, the blue incident light was often downconverted to red orange. Furthermore, a portion of the incident blue light was diffused into the large internal photonic glass borders, making them more visible.

Specimen 8 was the only specimen with sub-surface crazing. Crazing is an internal and/or surface stress fracturing process that results from thermal, hydration, vibration and/or photonic fluctuations [5,43]. Some CPP events were observed in reflections off of the internal planes of crazing and overlapping the crazed areas. Crazing did not appear to be a boundary to prevent CPP events. Yet, photonic properties did not always flow past the lines of crazing without distortion.

Early in this research, from certain angles of incidence, the CPP events were observed to align with the oval curvature of the exterior of specimen 8, with differentiated CPP events arranged similarly to the numbers around a clock. The rotations, shapes and polychroic

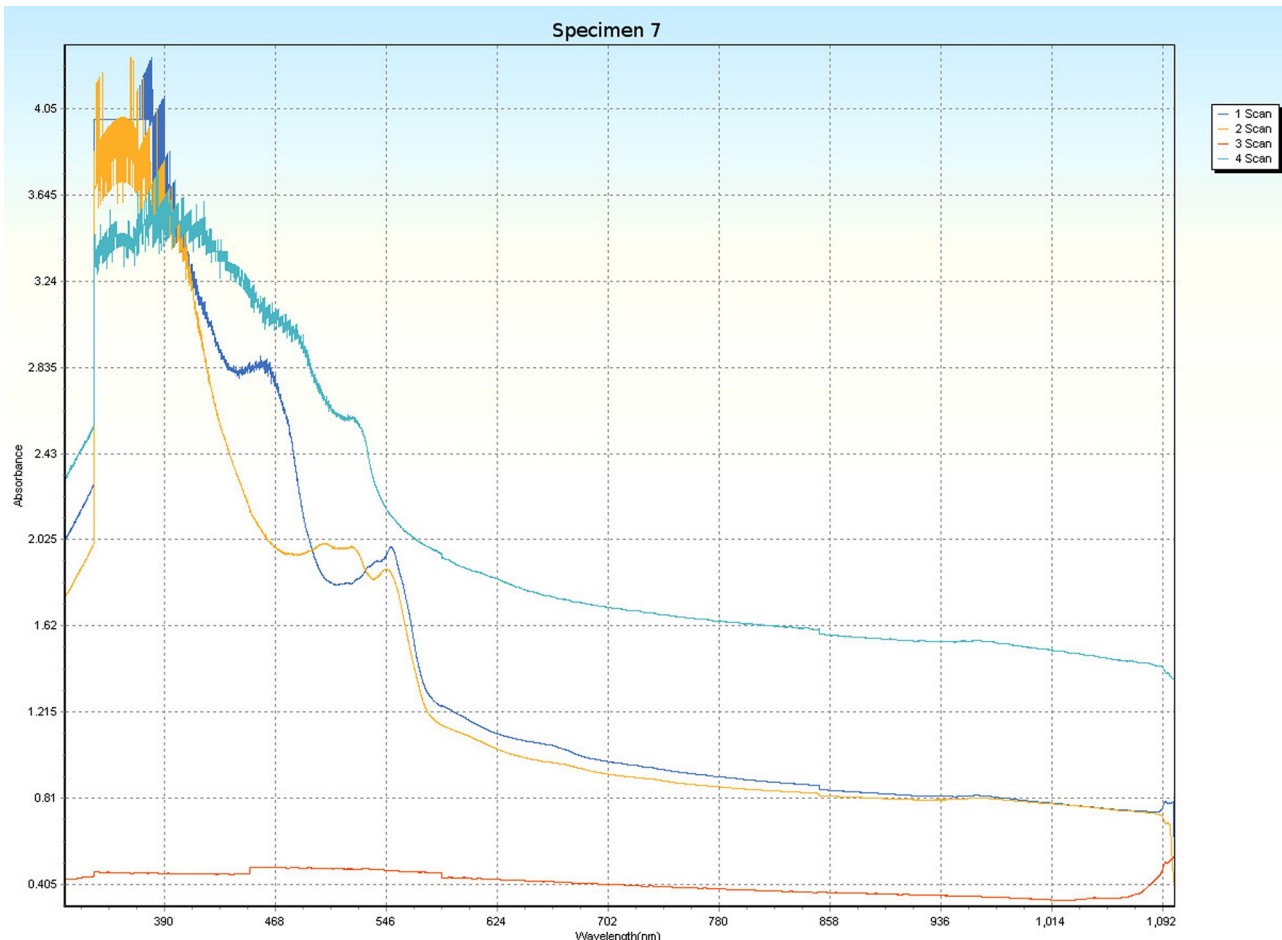

**Fig 21. Spectrophotometric scans of specimen 7.** a. top to bottom, b. bottom to top, c. side-to-side narrow, d. side-to-side long.

configurations of the CPP events displayed by this specimen made it one of the easiest to observe among the test specimens. However, many months of testing have witnessed progressive crazing. Crazing caused an increase in internal fractures and tiny surface flakes to dislodge from the specimen. Even with these physical issues, the specimen continued to display CPP events. However, the CPP events no longer assumed a clock-like formation and a decrease in the strength of clarity, coherence or polychroism developed.

## Conclusions

The present research identified and examined silicate materials, made of natural precious opal, that exhibited a previously undiscovered property of CPP. CPP enabled three-dimensional photonic control over visible wavelengths via demultiplexed diffraction, upconversion and/or downconversion of incident light with photonic coherence. The shape of the incident light source was propagated over three dimensions over multiple visible light frequencies. CPP events remained visible as each specimen was spatially manipulated under each incident visible light. Additionally, the specimens applied an unexpected axial rotational symmetry over the incident light. CPP and rotational properties were studied in isolation from exogenous thermal, photonic and electrical influences.

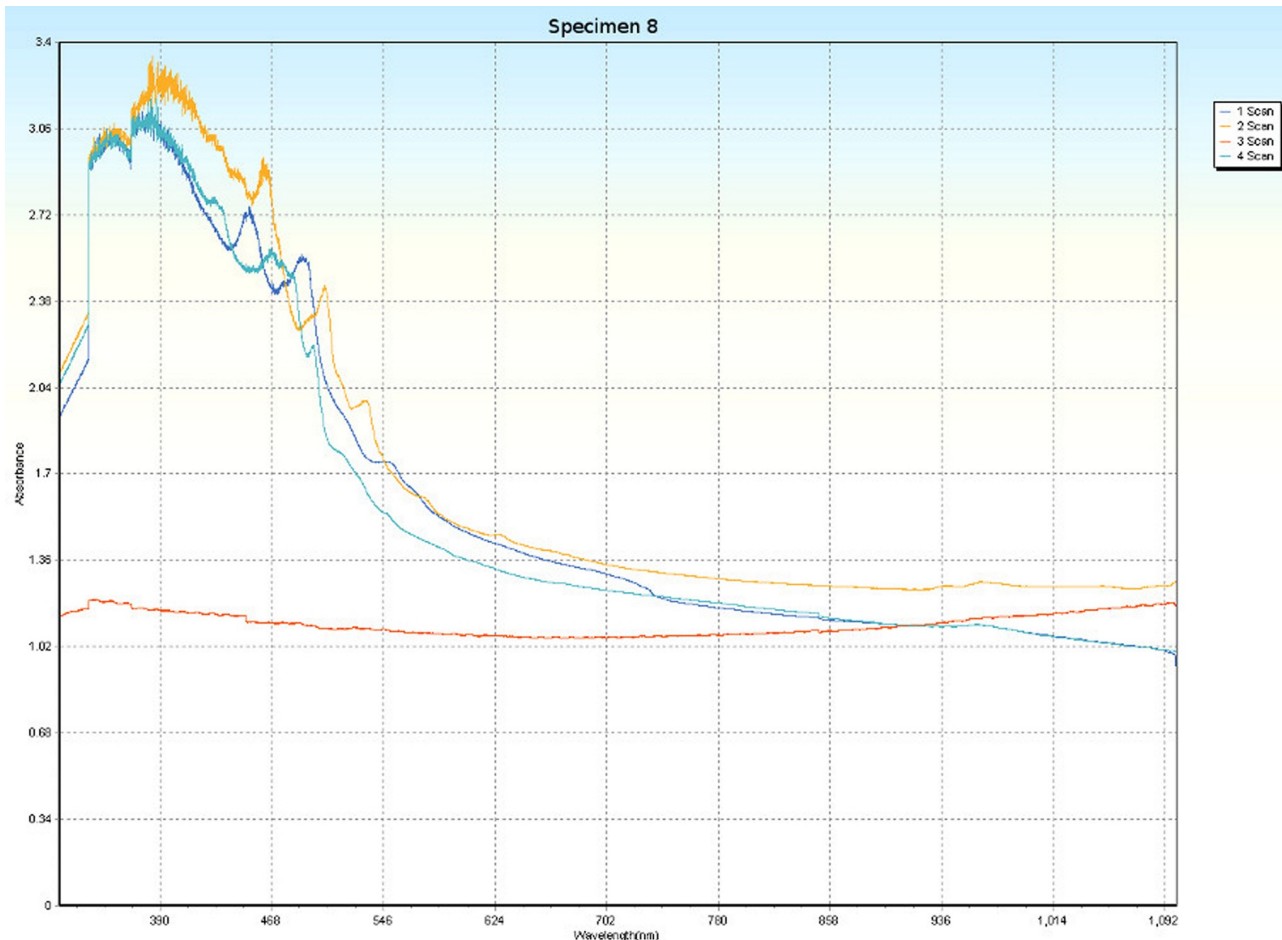

**Fig 22. Spectrophotometric scans of specimen 8.** a. top to bottom, b. bottom to top, c. side-to-side narrow, d. side-to-side long.

This research demonstrated 3-d photonic control via the CPP property. 3-d control was achieved with multiple specimens of natural opal. Observations of CPP occurred when the specimens and/or light sources were rotated relative to the viewer and over a variety of poly and monochromatic light sources (S1–S8 Figs).

CPP and other photonic properties were displayed. The photonic behaviors of each specimen were at least partially different from each of the others. The photonic control demonstrated by the specimens, particularly regarding CPP and PPOI rotation, contradicts common thought that precious opal is comprised of randomly oriented islands of pseudo-crystallinity which lack the ability to photonically behave in concert over a macroscopic volume (S1 Text General Observations) [1,3–6].

CPP was most prominent as a reflective property and much less prominent as a transmissive property. Therefore, CPP events were most strongly and frequently observed proximally rather than distally (Fig 1). CPP events visibly glided and rotated over proximal or distal surfaces as the specimens were moved or rotated relative to the incident lights and viewer.

Changes in the wavelength character of the incident light sources caused changes in the presence, strength and character of specimen photonic responses. Each of the photonic properties of each specimen was affected by the different incident light sources. The CPP materials propagated the shape of the incident photon source in multiple copies over multiple diffracted

wavelengths. Furthermore, each CPP specimen displayed photonic diffraction, upconversion and/or downconversion properties that varied when different mono and polychromatic light sources were applied. The CPP materials upconverted and downconverted incident mono-chromatic and polychromatic fiber optic light to generate polychromatic CPP events (S1–S8 Figs).

The specimens found to be the most unusual for each laser source were: specimen 2 for 650 nm laser, specimens 2 and 6 for 593.5 nm laser, specimens 5 and 6 for 532 nm laser, specimens 2 and 4 for 450 nm laser and specimens 7 and 8 for 405 nm laser (Table 3). The specimens found to be most unusual for each UV source were: specimens 5 and 7 for 375 nm photons, specimens 1, 2 and 4 for 307 nm photons and specimen 1 for 254 nm photons (Table 4). Specimen 3 was the only specimen that did not distinguish itself photonically for any of the mono-frequency examinations. Specimen 2 was the specimen that was most often identified as unusual in these mono-frequency examinations. Specimen 3 had mostly mildly positive CPP and specimen 2 had mostly moderately positive CPP. Hence, it may be possible to identify or anticipate the CPP property from the unusual interaction of a specimen with particular visible laser and/or UV wavelengths. While none of the specimens displayed CPP in response to UV light, play-of-color in response to UV light was observed for two specimens, which had not been documented previously.

Incident, reflected and transmitted PPOI were observed for almost all specimens (Fig 1). Incident PPOI almost always assumed the same shape, color(s) and orientation as the photon source. Transmitted PPOI was the most subject to distortion. While reflected and transmitted PPOI virtually always assumed the same shape as the photon source, they did not always propagate the incident color(s). Almost all reflected and transmitted PPOI displayed an axial rotation of 180° of the incident PPOI for all light sources (S1 Text General Observations). Axial rotation was more easily observable for asymmetrical light sources (Fig 3). A specular internal reflection off of the bottom of a specimen could not have caused the observed PPOI rotation [19]. Rotation occurred prior to incident PPOI reaching the distal side of the specimen since transmitted PPOI had also been rotated axially. Since opal is, overall, non-crystalline, this unexpected symmetry operation requires further study.

Specimen 3 was particularly sensitive to the angle of incident PPOI, relative to the viewer (S9 Fig). Regardless of the incident light chromaticity, specimen 3 had virtually the same angle-dependent response. Hence, specimen 3 upconverted and/or downconverted incident light often and reliably. When incident light was about 90° relative to the viewer, CPP and play-of-color events were mostly large visible wavelengths and when the incident light was at any other angle, CPP and non-CPP events were mostly small visible wavelengths (S1 Text General Observations). Specimens 3 and 5 had the highest maximum hydration percentages of the specimens, which did not correlate to the presence or strength of photonic properties.

The more complex spectrophotometric scans were likely to be scans of active PCZs that were caught in the act of displaying contra luz, play-of-color or CPP (S1–S8 Tables). The scan differences indicated that spectrophotometric results depended on the light path and orientation of a specimen, even when there were no visible intra-specimen differences in color tones or clarity. The spectrophotometric scans that displayed particularly strong peak systems (specimens 3, 5, 6, 7 and 8) gave insight into active play-of-color, contra luz or CPP events that was undocumented previously.

Polariscopically revealed strain and isogyre figures indicated that most of the present research specimens had uncommon microsphere organizations. Even so, all of the specimens were isotropic. Yet, the specimens that demonstrated the strongest presence of CPP were not strongly correlated to displaying the most unusual polariscopic responses.

Thermal conductivity and refractive index analyses confirmed the mineral species of the specimens to be natural precious opal. Hence, the specimens did not have sufficient PCZs to cause either test to indicate significant crystalline silicate composition. With increased instrumentation sensitivity, the slight differences in thermal conductivity readings among the specimens may be a future method to identify CPP materials.

Dichroscopy confirmed no dichroism, as expected. However, an asynchromatic response, showing dichroism of seven of the specimens when viewed through active contra luz events. This dichroscopic contra luz response had not been documented previously.

All of the specimens allowed 3-d photonic control to various extents. This research demonstrated that 3-d photonic output can be controlled by altering the relative angle of the specimen to incident light to viewer and by altering the chromaticity of incident light. As each specimen was rocked and rotated, relative to each of the incident light sources and viewer, intact CPP events were manifested (S1 Text General Observations). Some of these CPP events maintained the same chromaticity while gliding over the specimen and other CPP events changed colors as the relative angles changed. Yet, in all CPP and PPOI events, the shape of the original incident light was propagated with axial rotation. CPP and PPOI events were phenomenologically distinct properties and did not appear to be predictive or causative of each other.

Future research must identify means to analyze the specimens without shifting the microspheroids of the CPP specimens. A primary goal will be to manufacture materials that display refinements of the CPP property for 3-d photonic control.

## Supporting information

**S1 Fig. Photograph of specimen 1 under eight visible light sources.** a. CFL polychromatic, b. LED polychromatic, c. white fiber optic polychromatic, d. quadchromatic fiber optic polychromatic, e. blue fiber optic monochromatic, f. green fiber optic monochromatic, g. yellow fiber optic monochromatic, h. red fiber optic monochromatic. Three photos per specimen. (EPS)

**S2 Fig. Photograph of specimen 2 under eight visible light sources.** a. CFL polychromatic, b. LED polychromatic, c. white fiber optic polychromatic, d. quadchromatic fiber optic polychromatic, e. blue fiber optic monochromatic, f. green fiber optic monochromatic, g. yellow fiber optic monochromatic, h. red fiber optic monochromatic. Three photos per specimen. (EPS)

**S3 Fig. Photograph of specimen 3 under eight visible light sources.** a. CFL polychromatic, b. LED polychromatic, c. white fiber optic polychromatic, d. quadchromatic fiber optic polychromatic, e. blue fiber optic monochromatic, f. green fiber optic monochromatic, g. yellow fiber optic monochromatic, h. red fiber optic monochromatic. Three photos per specimen. (EPS)

**S4 Fig. Photograph of specimen 4 under eight visible light sources.** a. CFL polychromatic, b. LED polychromatic, c. white fiber optic polychromatic, d. quadchromatic fiber optic polychromatic, e. blue fiber optic monochromatic, f. green fiber optic monochromatic, g. yellow fiber optic monochromatic, h. red fiber optic monochromatic. Three photos per specimen. (EPS)

**S5 Fig. Photograph of specimen 5 under eight visible light sources.** a. CFL polychromatic, b. LED polychromatic, c. white fiber optic polychromatic, d. quadchromatic fiber optic

polychromatic, e. blue fiber optic monochromatic, f. green fiber optic monochromatic, g. yellow fiber optic monochromatic, h. red fiber optic monochromatic. Three photos per specimen. (EPS)

**S6 Fig. Photograph of specimen 6 under eight visible light sources.** a. CFL polychromatic, b. LED polychromatic, c. white fiber optic polychromatic, d. quadchromatic fiber optic polychromatic, e. blue fiber optic monochromatic, f. green fiber optic monochromatic, g. yellow fiber optic monochromatic, h. red fiber optic monochromatic. Three photos per specimen. (EPS)

**S7 Fig. Photograph of specimen 7 under eight visible light sources.** a. CFL polychromatic, b. LED polychromatic, c. white fiber optic polychromatic, d. quadchromatic fiber optic polychromatic, e. blue fiber optic monochromatic, f. green fiber optic monochromatic, g. yellow fiber optic monochromatic, h. red fiber optic monochromatic. Three photos per specimen. (EPS)

**S8 Fig. Photograph of specimen 8 under eight visible light sources.** a. CFL polychromatic, b. LED polychromatic, c. white fiber optic polychromatic, d. quadchromatic fiber optic polychromatic, e. blue fiber optic monochromatic, f. green fiber optic monochromatic, g. yellow fiber optic monochromatic, h. red fiber optic monochromatic. Three photos per specimen. (EPS)

**S9 Fig. Photographs of specimen 3 showing that its play-of-color palette is extremely sensitive to the angle of incident PPOI.** Both photos were taken under a CFL white light. PPOI rotational symmetry is clearly visible in the left photo. (EPS)

**S1 Table. Raw spectrophotometric data table for Specimen 1.** Spreadsheet generated graph at bottom of table. (PDF)

**S2 Table. Raw spectrophotometric data table for Specimen 2.** Spreadsheet generated graph at bottom of table. (PDF)

**S3 Table. Raw spectrophotometric data table for Specimen 3.** Spreadsheet generated graph at bottom of table. (PDF)

**S4 Table. Raw spectrophotometric data table for Specimen 4.** Spreadsheet generated graph at bottom of table. (PDF)

**S5 Table. Raw spectrophotometric data table for Specimen 5.** Spreadsheet generated graph at bottom of table. (PDF)

**S6 Table. Raw spectrophotometric data table for Specimen 6.** Spreadsheet generated graph at bottom of table. (PDF)

**S7 Table. Raw spectrophotometric data table for Specimen 7.** Spreadsheet generated graph at bottom of table. (PDF)

**S8 Table. Raw spectrophotometric data table for Specimen 8.** Spreadsheet generated graph at bottom of table.
(PDF)

**S1 Text. Extended experimentation details for all specimens.** Details regarding: General observations, compact fluorescent light, LED, fiber optic cluster, spectrophotometer and polariscope.
(DOC)

## Author Contributions

**Conceptualization:** Michelle R. Stem.

**Data curation:** Michelle R. Stem.

**Formal analysis:** Michelle R. Stem.

**Funding acquisition:** Michelle R. Stem.

**Investigation:** Michelle R. Stem.

**Methodology:** Michelle R. Stem.

**Project administration:** Michelle R. Stem.

**Resources:** Michelle R. Stem.

**Software:** Michelle R. Stem.

**Supervision:** Michelle R. Stem.

**Validation:** Michelle R. Stem.

**Visualization:** Michelle R. Stem.

**Writing – original draft:** Michelle R. Stem.

**Writing – review & editing:** Michelle R. Stem.

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
