## [Decision Letter · Decision Letter 0]

5 Sep 2019

[EXSCINDED]

PONE-D-19-20451

Coherent Poly Propagation Materials with 3-Dimensional Photonic Control over Visible Light

PLOS ONE

Dear Dr. Stem,

Thank you for submitting your manuscript to PLOS ONE. After careful consideration, we feel that it has merit but does not fully meet PLOS ONE’s publication criteria as it currently stands. Therefore, we invite you to submit a revised version of the manuscript that addresses the points raised during the review process.

We would appreciate receiving your revised manuscript by Oct 20 2019 11:59PM. To enhance the reproducibility of your results, we recommend that if applicable you deposit your laboratory protocols in protocols.io, where a protocol can be assigned its own identifier (DOI) such that it can be cited independently in the future. For instructions see: http://journals.plos.org/plosone/s/submission-guidelines#loc-laboratory-protocols

We look forward to receiving your revised manuscript.

Kind regards,

Zafar Ghouri

Academic Editor

PLOS ONE

Journal Requirements:

1. Please provide further details in your materials and methods section about the source of the opal samples, such that similar samples could be obtained by future researchers - this is in line with our reproducibility criterion for publishing, see https://journals.plos.org/plosone/s/criteria-for-publication#loc-3. Thank you for your attention to this query.

2. Thank you for inlcuding your competing interests statement; "The authors have declared that no competing interests exist."

We note that one or more of the authors are employed by a commercial company:Complete Consulting Services, LLC.

Additional Editor Comments (if provided):

Dear Michelle,

The reviewers has commented on your manuscript , PONE-D-19-20451 "Coherent Poly Propagation Materials with 3-Dimensional Photonic Control over Visible Light " and indicated that it is suitable for publication in PLOSE ONE after minor revision. I invite you to revise and resubmit you manuscript.

Below, please find the comments for your perusal.

With kind regards,

Zafar Khan Ghouri, Ph.D

Academic Editor

PLOSE ONE

Reviewers' comments:

Reviewer's Responses to Questions

**Comments to the Author**

1. Is the manuscript technically sound, and do the data support the conclusions?

Reviewer #1: Yes

Reviewer #2: Yes

2. Has the statistical analysis been performed appropriately and rigorously? 

Reviewer #1: Yes

Reviewer #2: Yes

3. Have the authors made all data underlying the findings in their manuscript fully available?

Reviewer #1: Yes

Reviewer #2: Yes

4. Is the manuscript presented in an intelligible fashion and written in standard English?

Reviewer #1: Yes

Reviewer #2: Yes

5. Review Comments to the Author

Reviewer #1: In this article (PONE-D-19-20451) the author studied in detail the coherent poly propagation effect over visible light by using Opal a nanosized silicate shapers. The study appears to be complete and the author has mentioned all the necessary arguments to satisfy the finding. Overall, this article appears interesting and comprehensive enough to be published in PLOS ONE. After addressing the following issue accept the manuscript.

The details of comments on this work are listed below:

1. Graphical figures that are shown in the manuscript looks like the cut paste of results generated by the analysis equipment which is not a good representation of data. These analysis equipment’s for example also can give the results is numerical data form which can be easily redrawn on other graphical software’s that are available like MS EXCEL. All the graphical data should be redrawn in Origin/ MS EXCEL software by the data generated by each figure.

2. Rearrange your supporting information and add a table of content for the supporting information.

Reviewer #2: 1. How the findings in the article shows advances in the field which will make difference with respect to scientific work already conducted previously.

2. Thermal conductivity method and procedure should be included in the article.

6. PLOS authors have the option to publish the peer review history of their article (what does this mean?). If published, this will include your full peer review and any attached files.

Reviewer #1: No

Reviewer #2: No

---

## [Author Response · Author response to Decision Letter 0]

18 Sep 2019

Dear Dr. Ghouri,

I am happy to submit the attached revised manuscript, “Coherent Poly Propagation Materials with 3-Dimensional Photonic Control over Visible Light”, to PLOS ONE for publication consideration as a research article.

As requested, details on the sources of the specimens were included in the materials and methods section. Also, a full response to the reviewers’ comments is attached.

Regarding funding, the only role of Complete Consulting Services, LLC was the provision of access to research equipment. There was no salary or any other considerations to the author as this is a very small research lab in which I serve an unpaid role.

Amended Funding Statement: Complete Consulting Services, LLC. provided support in the form of access to research equipment for the author (MRS) with no other financial or material contributions or considerations. The funder provided support in the form of access to research equipment only, but did not have any additional role in the study design, data collection and analysis, decision to publish, or preparation of the manuscript. The specific roles of the author is articulated in the ‘author contributions’ section.

No amendments to Competing Interests or Author Contributions Statements are needed. This does not alter our adherence to PLOS ONE policies on sharing data and materials.

During the process of making adjustments for formatting and per the recommendations of the reviewers, I corrected reference 4 to reflect a more recent work by the same author.

Thank you for helping me through this process. Please let me know if I missed anything.

Response to Reviewers

Reviewer #1:

1. Graphical figures that are shown in the manuscript looks like the cut paste of results generated by the analysis equipment which is not a good representation of data. These analysis equipment’s for example also can give the results is numerical data form which can be easily redrawn on other graphical software’s that are available like MS EXCEL. All the graphical data should be redrawn in Origin/ MS EXCEL software by the data generated by each figure.

Done. I gathered the raw spectrophotometric data in Excel format and generated an Excel graph of each specimen at the bottom of each specimen’s data. I included the Excel data in the supplementary section. Since the Excel graphs did not look as presentable, I used improved originals. I generated the original images via a better method and have replaced the lesser quality ones.

2. Rearrange your supporting information and add a table of content for the supporting information.

Done. I rearranged the supplementary file and added a table of contents.

Reviewer #2:

1. How the findings in the article shows advances in the field which will make difference with respect to scientific work already conducted previously.

Done. The introduction has been expanded to address this topic in better detail.

2. Thermal conductivity method and procedure should be included in the article.

Done. Thermal conductivity method, definition and procedure were added/expanded.

My sincere appreciation to each reviewer for generously giving their time and expertise to help improve my research manuscript. Thank you for your kind professionalism.

---

## [Decision Letter · Decision Letter 1]

27 Sep 2019

Coherent Poly Propagation Materials with 3-Dimensional Photonic Control over Visible Light

PONE-D-19-20451R1

Dear Dr. Stem,

We are pleased to inform you that your manuscript has been judged scientifically suitable for publication and will be formally accepted for publication once it complies with all outstanding technical requirements.

With kind regards,

Zafar Ghouri

Academic Editor

PLOS ONE

Additional Editor Comments (optional):

All the concerns have been addressed. Revised Manuscript has been accepted without further modification.

Reviewers' comments:

Reviewer's Responses to Questions

**Comments to the Author**

1. If the authors have adequately addressed your comments raised in a previous round of review and you feel that this manuscript is now acceptable for publication, you may indicate that here to bypass the “Comments to the Author” section, enter your conflict of interest statement in the “Confidential to Editor” section, and submit your "Accept" recommendation.

Reviewer #2: All comments have been addressed

2. Is the manuscript technically sound, and do the data support the conclusions?

Reviewer #2: Yes

3. Has the statistical analysis been performed appropriately and rigorously? 

Reviewer #2: Yes

4. Have the authors made all data underlying the findings in their manuscript fully available?

Reviewer #2: Yes

5. Is the manuscript presented in an intelligible fashion and written in standard English?

Reviewer #2: Yes

6. Review Comments to the Author

Reviewer #2: The author has successfully addressed the comments. Please accept the article for publication in the journal.

7. PLOS authors have the option to publish the peer review history of their article (what does this mean?). If published, this will include your full peer review and any attached files.

Reviewer #2: Yes: CAU

---

## [Editor Report · Acceptance letter]

2 Oct 2019

PONE-D-19-20451R1 

Coherent Poly Propagation Materials with 3-Dimensional Photonic Control over Visible Light 

Dear Dr. Stem:

I am pleased to inform you that your manuscript has been deemed suitable for publication in PLOS ONE. Congratulations! Your manuscript is now with our production department. 

With kind regards,

on behalf of

Dr. Zafar Ghouri 

Academic Editor

PLOS ONE